# Taking the GP Out of the Loop

**Mehul Bafna** [1]   **Siddhant Anand Jadhav** [1]   **David Sweet** [1]

## Abstract

Bayesian optimization (BO) has traditionally solved black-box problems where function evaluation is expensive and, therefore, observations are few. Recently, however, there has been growing interest in applying BO to problems where function evaluation is cheaper and observations are more plentiful. In this regime, scaling to many observations $N$ is impeded by Gaussian-process (GP) surrogates: GP hyperparameter fitting scales as $\mathcal{O}(N^3)$ (reduced to roughly $\mathcal{O}(N^2)$ in modern implementations), and it is repeated at every BO iteration. Many methods improve scaling at acquisition time, but hyperparameter fitting still scales poorly, making it the bottleneck. We propose Epistemic Nearest Neighbors (ENN), a lightweight alternative to GPs that estimates function values and uncertainty (epistemic and aleatoric) from $K$-nearest-neighbor observations. ENN scales as $\mathcal{O}(N)$ for both fitting and acquisition. Our BO method, TuRBO-ENN, replaces the GP surrogate in TuRBO with ENN and its Thompson-sampling acquisition with $\mathrm{UCB} = \mu(x) + \sigma(x)$. For the special case of noise-free problems, we can omit fitting altogether by replacing UCB with a non-dominated sort over $\mu(x)$ and $\sigma(x)$. We show empirically that TuRBO-ENN reduces proposal time (i.e., fitting time + acquisition time) by one to two orders of magnitude compared to TuRBO at up to 50,000 observations without sacrificing solution quality.

## 1. Introduction

Bayesian optimization (BO) is commonly used in settings where evaluations are expensive, such as A/B testing (days to weeks) (Quin et al., 2024; Sweet, 2023) and materials experiments (roughly 1 day) (Kotthoff et al., 2021). It has

---

[1]Department of Graduate Computer Science and Engineering, Yeshiva University, New York, NY 10033, USA. Correspondence to: David Sweet <david.sweet@yu.edu>.

*Proceedings of the 43$^{rd}$ International Conference on Machine Learning*, Seoul, South Korea. PMLR 306, 2026. Copyright 2026 by the author(s).

also been applied to simulation optimization problems in engineering, logistics, medicine, and other domains (Amaran et al., 2016). More recently, BO has been used in settings where evaluations are fast and can be run in parallel—for example, large-scale simulations in engineering design. In such cases, thousands of evaluations may be generated during a single optimization run (Daulton et al., 2022).

The proposal time for BO methods typically scales poorly with the number of observations, $N$, because proposals are generated by fitting and querying a Gaussian process (GP) surrogate. Modern, efficient implementations require $O(N^2)$ time per query. We refer to problems with large $N$ as *Bayesian optimization with many observations (BOMO)*, and present a method that reduces the proposal-time scaling to $O(N)$.

It is important to distinguish between BOMO and BO with many design parameters, i.e. high-dimensional Bayesian optimization (HDBO). Generally, we expect to need more observations to optimize more parameters since there are simply more possible designs to evaluate. This expectation is codified, for example, in Ax's (Meta, 2023) prescription to collect $2 \times D$ observations before fitting a surrogate (where $D$ is the number of design parameters, or *dimensions*). However, the number of observations necessary to locate a good design depends on aspects of a problem beyond just $D$. For example, (Wang et al., 2016) optimizes a one-billion-parameter analytical function with only 500 observations, while (Daulton et al., 2022) takes 1500 observations to optimize a challenging simulation problem with only 12 parameters.

This work focuses on BOMO. Specifically, we ask: **Can we make a state-of-the-art BO algorithm significantly faster on BOMO problems while producing comparable-quality solutions?** We are concerned mainly with scaling (with $N$) but we also report on wall time. Our approach is to strategically simplify TuRBO (Eriksson et al., 2019b), then compare solution quality, scaling, and running time.

We propose a novel $K$-nearest neighbors surrogate, Epistemic Nearest Neighbors (ENN), which estimates function values and uncertainty (both epistemic and aleatoric). We integrate this approach into TuRBO (Eriksson et al., 2019b) by replacing its GP surrogate and Thompson sampling acquisition method with ENN and UCB (Srinivas et al., 2010).

Figure 1 shows that GP-based BO methods exhibit approximately $O(N^2)$ running time, whereas alternative-surrogate methods, such as TuRBO-ENN, scale as $O(N)$. In addition to linear scaling, TuRBO-ENN achieves much lower absolute proposal times than the other methods.

**Contributions**   This paper makes the following contributions:

- We introduce *Epistemic Nearest Neighbors (ENN)*, a lightweight $K$-nearest-neighbor surrogate that estimates both function value and uncertainty (epistemic and aleatoric) with $O(N)$ fitting and querying time.

- We develop *TuRBO-ENN*, which replaces TuRBO's GP surrogate and Thompson sampling with ENN and UCB; for noise-free problems, we also describe a fitting-free variant based on non-dominated sorting over $(\mu(x), \sigma(x))$. Proposal (fitting + acquisition) time is $O(N)$.

- We empirically evaluate scaling and solution quality up to tens of thousands of observations in realistic simulation problems, showing one to two orders-of-magnitude reductions in proposal time while remaining competitive in optimization performance.

- We show that TuRBO-ENN is a *Pseudo-Bayesian Optimization* (PBO) method by verifying that its surrogate, uncertainty quantifier, and acquisition satisfy PBO conditions (Appendix C), which guarantee convergence.

In addition, we distribute TuRBO-ENN as an open-source, `pip`-installable Python package `ennbo` with source available at `https://github.com/yubo-research/enn`.

## 2. Background

### 2.1. Bayesian Optimization

A Bayesian optimizer proposes a design, $x \in [0,1]^D$, given some observations, $\mathcal{D} = \{(x_i, y_i, s_i)\}_{i=1}^N$,   $y_i = f(x_i) + s_i \varepsilon_i$, where $\varepsilon_i$ is a noise term and $s_i$ is the aleatoric uncertainty scale for observation $i$. A typical BO method consists of two components: a surrogate and an acquisition method. A surrogate is a model of $f(x)$ mapping a design, $x$, to both an estimate of $f(x)$, $\mu(x)$, and a measure of uncertainty in that estimate, $\sigma(x)$. An acquisition method determines the proposal, $x_p = \arg\max_x \alpha(\mu(x), \sigma(x))$, where the $\arg\max$ is found by numerical optimization (e.g., via BFGS (Meta, 2024a)) or by evaluating $\alpha(\cdot, \cdot)$ over a set of $x$ samples, for example, uniform in $[0,1]^D$ or following any number of sampling schemes (Kandasamy et al., 2018a; Eriksson et al., 2019b; Rashidi et al., 2024).

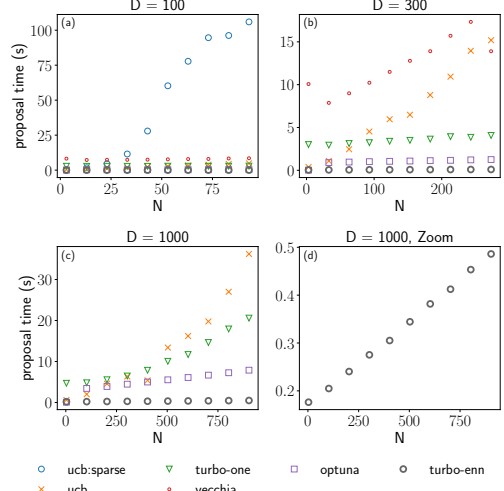

*Figure 1.* Mean proposal time (in seconds) versus number of observations ($N$) for several Bayesian optimization methods. We set $N = D$ in these runs. Subfigures show results for (a) $D = 100$, (b) $D = 300$, (c) $D = 1000$, and (d) a zoomed-in view of (c), averaged over many optimization runs (see Section 5). GP-based methods (`ucb` [Section 2.1.2] and `turbo-one`, original TuRBO method) scale approximately as $O(N^2)$, while `optuna`, which uses a Parzen estimator, `vecchia`, a nearest-neighbor GP method, and our method (`turbo-enn`) scale linearly in $N$. Scaling behavior for `ucb:sparse`, which uses a sparse GP, is not simple. Results are averaged over 12 functions × 10 BO runs/function = 120 runs for each optimization method. The functions, {`ackley`, `rastrigin`, `sphere`, `trid`, `booth`, `mccormick`, `dixonprice`, `rosenbrock`, `dejong5`, `easom`, `branin`, `stybtang`}, consist of two functions each from the six categories of optimizer test function in (Surjanovic & Bingham, 2013). The various methods are discussed in more detail in Section 3.

#### 2.1.1. SURROGATE

The usual BO surrogate is a Gaussian process (Rasmussen & Williams, 2006). For simplicity, assume homoscedastic observation noise,

$$y_i = f(x_i) + \varepsilon_i, \qquad \varepsilon_i \sim \mathcal{N}(0, \sigma_0^2).$$

Given observations $\mathcal{D} = \{(x_i, y_i)\}_{i=1}^N$, the GP posterior at a new point $x$ has mean and variance

$$A \equiv K(X, X) + \sigma_0^2 I, \tag{1}$$

$$\mu(x) = K(X, x)^\top A^{-1} y, \tag{2}$$

$$\sigma^2(x) = K(x, x) - K(X, x)^\top A^{-1} K(X, x). \tag{3}$$

where $\sigma_0$ is an inferred aleatoric uncertainty. The $N \times N$ kernel matrix, $K(X, X)$, has elements $K(X, X)_{ij} = k(x_i, x_j)$, where $k(\cdot, \cdot)$ is a kernel function, e.g., a squared exponential $k(x_i, x_j) = \exp(-\|x_i - x_j\|^2/(2\lambda))$, although others are common, too (Rasmussen & Williams,

2006). Similarly, the kernel vector $K(X, x) \in \mathbb{R}^N$ has entries $K(X, x)_i = k(x_i, x)$. The kernel matrix is the pairwise covariance between all observations, and the kernel vector is the covariance between the query point $x$ and the observations.

Constructing the $N \times N$ kernel matrix $K(X, X)$ takes $N(N - 1)/2$ evaluations of $k(\cdot, \cdot)$, which is $O(N^2)$. Exact GP training then requires solving linear systems involving $K(X, X)$ (typically via a Cholesky factorization), which is $O(N^3)$ in general; modern iterative solvers can reduce key GP computations to roughly $O(N^2)$ (Gardner et al., 2018).

**Hyperparameter fitting** The hyperparameters, $\lambda$ and $\sigma_0$, the kernel length-scale and noise level, are typically chosen to maximize the marginal log-likelihood of the observations, $\mathcal{D}$, by a numerical optimizer such as SGD (Eriksson et al., 2019b) or BFGS (Meta, 2024a). When optimizing, one needs to reconstruct $K$ for each hyperparameter proposed by the optimizer.

### 2.1.2. ACQUISITION METHOD

There are many acquisition methods in the literature. Three common ones are:

**Upper Confidence Bound (UCB)** $x_p = \arg\max_x \big[\mu(x) + \beta\sigma(x)\big]$, where $\beta$ is a constant. The first term encourages exploitation of $\mathcal{D}$, i.e. biasing $x_p$ towards a design that is expected to work well, while the second term encourages exploration of the design space so as to collect new observations that will improve future surrogates.

**Expected Improvement (EI)** $x_p = \arg\max_x \mathbb{E}[\max\{0, y(x) - y_*\}]$, where $y_* = \max y_m$, and the expectation is taken over $y(x) \sim \mathcal{N}(\mu(x), \sigma^2(x))$.

**Thompson Sampling (TS)** $x_p = \arg\max_x y(x)$, where $y(x)$ is a joint sample from the GP at a set of $x$ values (Kandasamy et al., 2018a). (A joint sample modifies equation 3 to account for the covariance between each $x$ that is being sampled.)

## 3. Related Work

Approaches to scaling BO to many observations include acceleration of exact GP computations, trust regions, and various alternative surrogates. Our method combines a trust region with a novel alternative surrogate, ENN. We briefly review various approaches below.

**Blackbox Matrix-Matrix Multiplication** A conjugate-gradient algorithm replaces the inversion of $K(X, X)$ in equation equation 3 with a sequence of matrix multiplies, reducing the hyperparameter-fitting time complexity of a GP from $O(N^3)$ to $O(N^2)$ (Gardner et al., 2018). A Lanczos algorithm can speed up GP posterior sampling (e.g., used in

Thompson sampling) to constant-in-$N$ (Pleiss et al., 2018).

**Trust Region BO** The TuRBO algorithm (Eriksson et al., 2019b) reduces wall-clock time in two ways: (i) It occasionally restarts, discarding all previous observations, resetting $N$ to 0. (ii) It restricts Thompson samples to within a trust region, a small subset of the overall design space where good designs are most likely, thus avoiding needless evaluations elsewhere. Our method replaces the GP surrogate in TuRBO with ENN.

**Sparsity** Sparse GP methods (Titsias, 2009) replace observations in the covariance matrix with $M$ summary (inducing) points, reducing the training complexity of GP fitting to $O(NM^2)$ and inference to $O(NM)$. Optimally, $M$ increases only slowly with $N$ (Burt et al., 2020), although the recommended setting for a popular BO library (Meta, 2024a) is $M = 0.25N$, which results in $O(N^2)$ inference. Fitting requires choosing inducing points (various approaches exist (Moss et al., 2023)) and can be computationally intensive, since the variational loss is more complex than the exact GP likelihood. Figure 1 shows a sparse GP-based method displaying a complex proposal time behavior with $N$.

**Modeling $p_*(x)$** An open-source optimizer, Optuna (Akiba et al., 2019; Optuna, 2025), does not model $f(x)$. Instead, it models $p_*(x) = P\{x = \arg\max_x f(x)\}$. The model is a Parzen estimator, a linear combination of functions of the observations, which has $O(N)$ query time. Optuna uses a modified EI-based acquisition method (Watanabe, 2023).

**Parametric surrogates** Other methods of scaling to large $N$ replace the GP with a neural network (DNGO, (Snoek et al., 2015)) or a random forest (SMAC, (Hutter et al., 2011)). While fitting a neural network or random forest scales as $O(N)$, the fitting processes are complex and introduce many tunable hyperparameters. Query time depends on the model architecture and is independent of $N$. Due to the fitting complexity, we do not compare to these methods in this paper.

**Nearest-neighbors** Vecchia GP methods (Jimenez & Katzfuss, 2023; Jimenez, 2025) and others (Gramacy & Apley, 2015; Wu et al., 2022) condition GP inference only on $M$ nearest neighbors. Fitting and inference scale as $O(NM)$. One Vecchia GP-based optimization method explored in (Jimenez & Katzfuss, 2023) combines nearest-neighbor lookups with TuRBO's trust region sampling. Our approach is similar in that it uses nearest-neighbor lookups and a trust region, but our ENN estimates are constructed from simpler linear combinations of observations (see Section 4), resulting in a significant speedup, see Figure 1.

Note that none of the alternative (non-GP) surrogates discussed above, except for ENN, supports principled uncertainty quantification.

Chen & Lam (2025) recently introduced *Pseudo-Bayesian*

*Optimization (PBO)*, which establishes convergence guarantees for any method whose surrogate, uncertainty quantifier, and acquisition satisfy certain conditions. Appendix C shows that TuRBO-ENN satisfies these conditions and, therefore, qualifies as a PBO method, inheriting the associated convergence guarantees.

## 4. Epistemic Nearest Neighbors

### 4.1. ENN Surrogate

A surrogate maps designs or parameters, $x$, to measurements, $y = f(x) + s\varepsilon$, where $s$ is the aleatoric uncertainty in $y$. A triple, $(x, y, s)$ is called an observation. A data set $\mathcal{D}$ is a collection of observations, $(x_m, y_m, s_m) \in \mathcal{D}$.

We define our ENN surrogate by four properties. For a query point, $x$, given $\mathcal{D}$,

- **Independence**: Each observation, $(x_m, y_m, s_m) \in \mathcal{D}$ is an independent estimate of $f(x)$.

- **Conditional mean**: $\mu(x \mid x_m, y_m, s_m) = y_m$.

- **Conditional aleatoric variance**:
  $\sigma_a^2(x \mid x_m, y_m, s_m) = s_0^2 + s_m^2$

- **Conditional epistemic variance**:
  $\sigma_e^2(x \mid x_m, y_m, s_m) = c_e d^2(x, x_m)$

where $d(x, x_m)$ denotes the (Euclidean) distance from $x$ to $x_m$, and $s_0$ and $c_e$ are hyperparameters. Notice that if $s_m = 0$, then aleatoric uncertainty is purely inferred ($s_0^2$) and homoscedastic.

Precisely speaking, we *treat* the estimates as independent for tractability. Equating epistemic variance to squared distance from the measurement, $x_m$, captures the intuition that similar designs will have similar evaluations, $f(x)$.

**Combining estimates**  For a query point, $x$, we combine the mean estimates from each of its $K$ nearest neighbors using the linear combination with minimum variance, the precision-weighted average (Cochran, 1954). Defining $\sigma^2(x|x_i, y_i, s_i) = \sigma_a^2(x|x_i, y_i, s_i) + \sigma_e^2(x|x_i, y_i, s_i)$, we write

$$\mu(x) = \frac{\sum_i^K \sigma^{-2}(x \mid x_i, y_i, s_i)\mu(x \mid x_i, y_i, s_i)}{\sum_i^K \sigma^{-2}(x \mid x_i, y_i, s_i)}$$

and similarly for the aleatoric variance

$$\sigma_a^2(x) = \frac{\sum_i^K \sigma^{-2}(x \mid x_i, y_i, s_i)\sigma_a^2(x \mid x_i, y_i, s_i)}{\sum_i^K \sigma^{-2}(x \mid x_i, y_i, s_i)}$$

Finally, we estimate epistemic variance of $\mu(x)$:

$$\sigma_e^2(x) = \mathrm{Var}[\mu(x)] = \frac{1}{\sum_i^K \sigma^{-2}(x \mid x_i, y_i, s_i)}$$

**Computational cost**  Finding the $K$ nearest neighbors requires evaluating $d(x, x_i)$ for all $N$ observations, at a cost of $O(N \ln K)$ time per query (using a max heap (Cormen et al., 2009)). Treating the observations as independent relieves us from calculating the $O(N^2)$ pairwise covariances between observations as in the calculation of $K(x_m, x)$ in equation 3. Our implementation uses the Python module Faiss (Meta, 2024b) to find the $K$ nearest neighbors.

**Hyperparameters**  The hyperparameters $s_0$ and $c_e$ are fit by Type-II MLE optimization of an LOOCV [leave-one-out cross validation (Hastie et al., 2009)] average pseudolikelihood (Rasmussen & Williams, 2006).

$$\log \mathcal{L}(\theta) = \frac{1}{N} \sum_{n=1}^{N} \ell_{-n}(\theta)$$

where $\theta = \{s_0, c_e\}$, and

$$\ell_{-n}(\theta) = -\frac{1}{2}\Big[ \log\big(2\pi\sigma_{-n}(x_n; \theta)^2\big)$$
$$+ \frac{\big(y_n - \mu_{-n}(x_n; \theta)\big)^2}{\sigma_{-n}(x_n; \theta)^2} \Big].$$

where $(\mu_{-n}(x_n; \theta), \sigma_{-n}(x_n; \theta))$ are LOO ENN estimates.

Each LOO estimate takes $O(N \ln K)$ time. Forming $N$ of them would, thus, take $O(N^2 \ln K)$. To avoid introducing an $\sim N^2$ computation, we approximate the average pseudo-likelihood with a fixed-size, $P$, subsample ($S_P$) of the $N$ observations.

$$\widehat{\log \mathcal{L}}(\theta) = \frac{1}{P} \sum_{n \in S_P} \ell_{-n}(\theta).$$

To calculate $\widehat{\log \mathcal{L}}(\theta)$ we need to produce LOO ENN estimates for $P$ points, taking only $O(PN \ln K)$ calculations.

We justify a constant $P$ by noting that we only require our likelihood estimate to be precise enough to support a decision about which hyperparameters to use. If the variance $\mathrm{Var}[\ell_{-n}(\theta)] = \sigma_\ell^2(\theta)$, then

$$\mathrm{Var}\Big[\widehat{\log \mathcal{L}}(\theta)\Big] = \frac{1}{P}\sigma_\ell^2(\theta).$$

Hence, fixing $P$ fixes the precision (inverse variance) of the likelihood estimate (independently of $N$).

To further limit the amount of computation devoted to hyperparameter tuning, we fix $K$ to a single value for the entirety of all BO runs. We find that $K = 10$ gives good performance and leave it fixed there throughout this paper. Appendix A examines ENN's and TuRBO-ENN's dependence on $K$ and $P$.

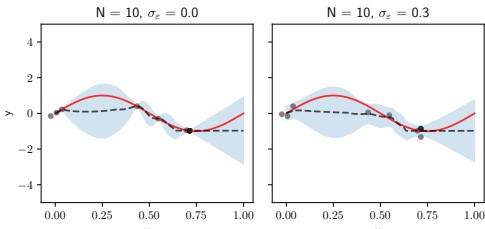

*Figure 2.* Epistemic nearest neighbors (ENN) surrogate for two noise levels, $\sigma_\varepsilon$. The dashed line shows $\mu(x)$ and the shaded region is $\pm 2\sigma(x)$. The solid red line is the function being estimated, $f(x) = \sin(2\pi x)$.

Figure 2 depicts the ENN surrogate for $f(x) = \sin(2\pi x)$ with various noise levels. Note that, in general, the ENN estimate (dashed line) may or may not pass through an observation. In particular, the three right-most observations all have the same $x$ value but different $y$ values due to noise, yet ENN provides a single mean estimate at that $x$ value; ENN is an estimator, not an interpolator.

### 4.2. Acquisition

We follow a modified version of the acquisition method of TuRBO (Eriksson et al., 2019b). Candidate generation is the same in both cases: we center a trust region $T \subseteq [0,1]^D$ at an incumbent point $x_0$, then generate many candidates $\{x_c\} \subset T$ via RAASP sampling (Rashidi et al., 2024; Regis & Shoemaker, 2013). The difference is how we (i) choose the incumbent and (ii) choose the next arm from the candidates.

#### Noise-free acquisition

- **Incumbent**: set $x_0 \in \arg\max_{x \in \{x_1, \ldots, x_N\}} y(x)$.

- **Arm selection**: Compute $(\mu(x_c), \sigma(x_c))$ for candidates and run a non-dominated sort (NDS) (Buzdalov & Shalyto, 2014; De Ath et al., 2021) over the two objectives $\mu(x_c)$ and $\sigma(x_c)$. NDS produces a Pareto front of candidates, from which we randomly sample the proposed arm, $x_{\text{arm}}$.

This differs from the original TuRBO, which selects $x_{\text{arm}}$ via Thompson sampling: $x_{\text{arm}} = \arg\max_{x_c} \{y(x_c)\}$ where $\{y(x_c)\}$ is a joint GP sample over the candidates.

For noise-free problems (common in simulation (Santner et al., 2019)), we omit hyperparameter fitting altogether, as ENN hyperparameter fitting only serves to calibrate uncertainty to match $\mu(x)$ (i.e., to determine $c_e$) and to infer the noise level (which is known, *a priori*, for noise-free problems, to be $s_0 = 0$). But NDS never directly compares $\sigma(x)$ to $\mu(x)$. All comparisons in the sort are among $\sigma(x)$ values or among $\mu(x)$ values separately (Buzdalov & Shalyto, 2014). Thus, fitting the hyperparameters for noise-free

problems at best wastes compute time, and, at worst, risks introducing hyperparameter estimation error (i.e., setting $s_0 \neq 0$).

#### Noisy acquisition

- **Fit**: Fit ENN hyperparameters $(s_0, c_e)$ by maximizing the subsampled LOO pseudolikelihood $\widehat{\log \mathcal{L}(\theta)}$ (Section 4).

- **Incumbent**: Eriksson et al. (2019b) recommends selecting the maximizer of the denoised observations, $x_0 = x_{\arg\max_i \mu(x_i)}$. We avoid computing $\mu(x_i)$ for all $N$ observations since that would take $O(N^2 \ln K)$ time. Instead, we first subset to the top-$K$ observed $y_i$, then set $x_0 = x_{\arg\max_{i \in I_{\text{top}}} \mu(x_i)}$ where $I_{\text{top}}$ are the indices of those top-$K$ observations.

- **Arm selection**: Compute $(\mu(x_c), \sigma(x_c))$ for candidates and select $x_{\text{arm}} = \arg\max_{x_c} UCB(x_c)$ where $UCB(x) = \mu(x) + \sigma(x)$ (we take $\beta = 1$).

Algorithm 1 summarizes both variants. An implementation of TuRBO-ENN, along with the original TuRBO, is available in the `pip`-installable Python library, `ennbo` with source and examples on GitHub at `github.com/yubo-research/enn`.

For the exact, original specifications of $T \leftarrow T(x_0, \ell)$, $\text{RAASP}(T)$ and $\ell \leftarrow \text{update}(\ell)$, please see (Eriksson et al., 2019b;a).

## 5. Numerical Experiments

We benchmark TuRBO-ENN by optimizing five noisy simulations both with *natural* and *frozen* noise, which we define subsequently (Section 5.2). Additionally, we compare ENN to other surrogates (described in Section 3) on pure fitting tasks.

### 5.1. Simulators

**LunarLander-v3** A 2D physics simulation (Farama, 2024) where the goal is to safely land a spacecraft at a designated stop using continuous engine thrusters while managing fuel consumption and orientation. The controller is the 12D heuristic, non-differentiable controller presented in (Eriksson et al., 2019b). The simulators for this task and the next two are implemented in the Gymnasium (fmr. OpenAI gym) Python package (Brockman et al., 2016).

**Hopper-v5** A 2D physics simulation (Farama, 2024) where the goal is to safely hop a one-legged robot forward. The controller is a 34D linear mapping from states to actions with hard clamping of action values, similar to (Mania et al., 2018).

**Algorithm 1** TuRBO-ENN with two acquisition variants: noise-free (ENN+NDS, no fitting) and noisy (fit ENN + UCB)

---

1: **Input:** objective $f : [0, 1]^D \to \mathbb{R}$; rounds $R$; $K$; $P$; `noise_free` (bool); trust region lengthscale, $\ell$
2: **Initialize:** LHD sample $\{x_i\}_{i=1}^{N_{\text{init}}}$, evaluate $y_i = f(x_i)$, set $\mathcal{D} \leftarrow \{(x_i, y_i)\}$; initialize trust region $T$
3: **for** $t = 1$ to $R$ **do**
4:    **if** `noise_free` **then**
5:       $s_0 \leftarrow 0, c_e \leftarrow 1$
6:    **else**
7:       $(s_0, c_e) \leftarrow \arg\max_{s_0, c_e} \ \text{LOOCV}_P(s_0, c_e)$
8:    **end if**
9:    **if** `noise_free` **then**
10:      $x_0 \leftarrow x_{\arg\max_i y_i}$
11:    **else**
12:      $I_{\text{top}} \leftarrow \text{topK}(\{y_i\}; K)$
13:      $x_0 \leftarrow x_{\arg\max_{i \in I_{\text{top}}} \mu(x_i)}$
14:    **end if**
15:    $T \leftarrow T(x_0, \ell)$
16:    $\{x_c\} \leftarrow \text{RAASP}(T)$
17:    $(\mu_c, \sigma_c) \leftarrow \text{ENN}(\{x_c\}; \mathcal{D}, s_0, c_e)$
18:    **if** `noise_free` **then**
19:      $x_{\text{arm}} \sim \text{Front}_1(\text{NDS}(\{x_c\}; \mu_c, \sigma_c))$
20:    **else**
21:      $x_{\text{arm}} \leftarrow \arg\max_{x_c}[\mu_c + \sigma_c]$
22:    **end if**
23:    Evaluate $y_{\text{arm}} \leftarrow f(x_{\text{arm}})$ and append $(x_{\text{arm}}, y_{\text{arm}})$ to $\mathcal{D}$
24:    $\ell \leftarrow \text{update}(\ell)$
25: **end for**
26: **Output:** $x_{\text{best}} \in \arg\max_{(x_i, y_i) \in \mathcal{D}} y_i$

---

**BipedalWalker-v3** A 2D physics simulation (Farama, 2024) where the goal is to navigate across randomly generated rough terrain using a two-legged robot. The controller is a 16D heuristic, non-differentiable controller designed interactively with Cursor (Cursor), GPT-5.2 (OpenAI), and Claude Opus 4.5 (Anthropic). Code for this controller is available at <ANONYMOUS>.

**Push-v5** A 2D physics simulation where the goal is to push two objects using two robot hands. The controller is the 14D heuristic, non-differentiable controller studied in (Wang et al., 2017) and (Eriksson et al., 2019b).

**LASSO-DNA** A 180D hyperparameter optimization (HPO) problem from LASSOBench (Šehić et al., 2022). The objective is to minimize a validation error over 180 LASSO weights. This HPO problem is similar to a simulation optimization problem in that the evaluation time is short and many observations may be needed to find an ac-

ceptable solution. In (Šehić et al., 2022), TuRBO gave the best performance of the optimizers tested on this problem. (NB: We treat this as a maximization problem by flipping the sign of the objective.)

## 5.2. Noise Handling

The simulators are driven, in part, by a seeded random number generator such that multiple runs with the same controller and different seed values will produce different episode returns (i.e., the objective values). We take two approaches to handling noise, natural and frozen, and produce comparisons for each.

### 5.2.1. NATURAL NOISE

Natural noise models a setting where one cannot control the noise source with a seed. Some simulation environments might be unavoidably non-deterministic (Isaac Lab Project Developers), and, of course, any physical system would produce uncontrolled noise.

For this style, we use a different seed for *every* episode of the simulator throughout the entire optimization run. We reproduce the same seed sequence for each optimization method to make their runs more comparable. Thus, to a single optimizer, the simulator looks noisy, yet each optimizer sees the same noise.

To evaluate an optimizer, at each round we ask it for its pick of a "best" parameter setting. With that setting, we simulate several (e.g., 10) times with a fixed set of evaluation seeds (distinct from the seeds used during optimization) and record the mean episode return, $r_{\text{passive}}$. The intention of $r_{\text{passive}}$ is to estimate the expected episode return we would achieve were we to stop the optimizer and accept its best parameter setting for future, prolonged use of the controller. Note that the optimizer's pick of "best setting" is sensitive to noise and may not yield the best expected future episode return. Thus, the natural-noise plots are not generally monotonically increasing. Note that the optimizer never sees $r_{\text{passive}}$. In our figures, the number of passive evaluation seeds used to compute $r_{\text{passive}}$ is `num_denoise_passive`.

To pick the best parameter setting, (Eriksson et al., 2019b) recommends selecting the observation, $(x_i, y_i)$, with the largest $\mu(x_i)$ value, as given by the surrogate. We follow this approach for TuRBO, but modify it for TuRBO-ENN. In the latter case, we first subset to the top $K$ observations by $y_i$, then find $\arg\max_i \mu(x_i)$ of this $K$-sized set. Preprocessing by subsetting reduces the time scaling of picking the max from $O(N^2 \ln K)$ to $O(NK \ln K)$.

### 5.2.2. FROZEN NOISE

Frozen noise (Kim et al., 2003) converts a noisy simulator into a noise-free objective function. To implement frozen

noise, we produce a fixed set, $S$, of seeds at the outset of the optimization run. To evaluate a design, we run the simulator once for each seed in $S$ and return to the optimizer the mean episode return over all seeds. Noise-free problems tend to be easier to solve, as we'll see in the figures below, and frozen noise is an approach commonly employed in BO and RL literature (Kim et al., 2003; Eriksson et al., 2019b; Salimans et al., 2017; Schulman et al., 2017). In our figures, the number of seeds in the seed set is named `num_denoise_obs`.

### 5.3. Results

We label our method `turbo-enn`; it uses the noisy ENN+UCB variant for natural-noise experiments and the noise-free ENN+NDS variant for frozen-noise experiments. We label the original TuRBO `turbo-one` (one trust region). We include an ablation, `turbo-zero`, which proposes arms by uniformly sampling from TuRBO's RAASP candidate set without a surrogate. We also compare to Optuna (`optuna`) (Optuna, 2025), CMA-ES (`cma`) (Hansen, 2023), SMAC (`smac`) (Lindauer et al., 2016) , DNGO (`dngo`) (Snoek et al., 2015) , and GP-UCB (`ucb`) (Meta, 2024a), as described in Section 3, as well as a `random` baseline that samples arms uniformly in $[0, 1]^D$. We repeat each optimization run 30 times with different seed sets and plot the mean $\pm$ two standard errors.

Across tasks, `turbo-enn` matches `turbo-one` in optimization quality while reducing total proposal time by about one to two orders of magnitude (Table 1), with larger speedups at larger $N$. Our largest runs use up to $N = 50,000$ observations. In the figures, $y_{\text{best}}$ denotes either the mean objective on the passive evaluation seeds (natural noise, left subplots) or the best objective seen so far (frozen noise), and $N$ is the number of observations collected. CMA-ES requires `num_arms` > 1 and is therefore omitted from sequential natural-noise runs (`num_arms`=1). Some methods exceeded a 5-hour per-run budget for cumulative proposal time and are omitted, appearing as `>5hr` in Table 1.

Natural noise is generally more difficult for all algorithms than frozen noise. Speedup, the ratio of the proposal time of `turbo-one` to the proposal time of `turbo-enn`, ranges from about $8.5\times$ to about $151\times$ across the settings in Table 1 (column-wise ratios of reported proposal times). Speedup is problem-dependent, but it is generally larger for larger $N$ and for frozen noise. Note that the frozen noise algorithm omits hyperparameter fitting, reducing computation time. The surrogate-free algorithm `turbo-zero` performs well on frozen-noise problems but is rarely the best, and it suffers on natural-noise problems.

In our work we dedicate a single CPU core to each run to create meaningful comparisons of wall time. Times reported

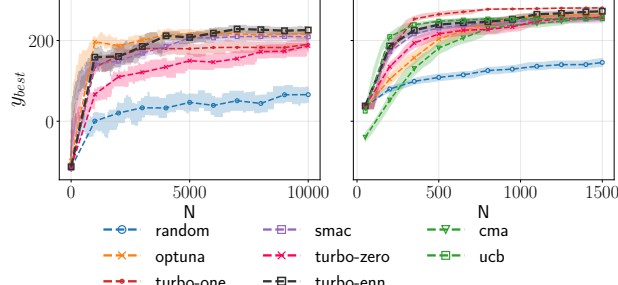

*Figure 3.* LunarLander-v3, $D = 12$, using the controller presented in (Eriksson et al., 2019b). Left: Natural noise. `num_arms` = 1, `num_denoise_passive` = 30. Right: Frozen noise. `num_arms` = 50, `num_denoise_obs` = 50.

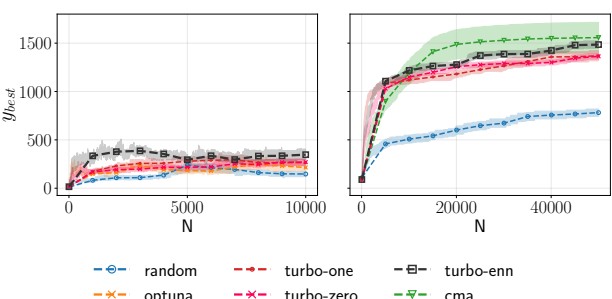

*Figure 4.* Hopper-v5, $D = 34$, using a linear controller, similar to (Mania et al., 2018). Left: Natural noise. `num_arms` = 1, `num_denoise_passive` = 10. Right: Frozen noise. `num_arms` = 50, `num_denoise_obs` = 10.

for individual methods in other papers may use different (typically more) resources per run (e.g., 32 cores (Hutter et al., 2011) or GPU (Eriksson et al., 2019b)).

### 5.4. Pure Fitting Tasks

To isolate surrogate *fitting* from BO acquisition, we time fits to synthetic training sets and score predictions on a held-out test set (Section 3 for model definitions). Figure 8 shows one representative benchmark (Ackley, $D$=10, $N$=1...$10^6$ training points, $N_{test}$=100 test points); Appendix B reports the same protocol across training sizes $N$ and additional test functions. ENN is orders of magnitude faster than several baselines when runs finish within a 5-hour wall-clock timeout, while its test-set log-likelihood ranks near the median or better among surrogates (cf. Figure 17 in the appendix). Some points are ommitted for small $N$ because implementations of the given surrogates do not support small $N$. Some points are ommitted for large $N$ because fitting exceeded a 5-hour wall-clock limit.

## 6. Limitations and Future Work

Improving scaling from $O(N^2)$ to $O(N)$ enables BO to handle many more observations. We wonder whether scal-

*Table 1.* Total proposal time (seconds) for each optimization method, summed over the full optimization run, for natural-noise (left subplots) and frozen-noise (right subplots) experiments. Dashes indicate that the method was not run for that setting. Speedup is computed as the ratio of the proposal time of `turbo-one` to the proposal time of `turbo-enn`. We set a timeout for a single run at 5 hours of cumulative proposal time.

| Method / field | LunarLander-v3 | | Hopper-v5 | | BipedalWalker-v3 | | Push-v5 | | LASSO-DNA | |
|---|---|---|---|---|---|---|---|---|---|---|
| $D$ | 12 | | 34 | | 16 | | 14 | | 180 | |
| $N$ | 10000 | 1500 | 10000 | 50000 | 10000 | 5000 | 10000 | 15000 | 1000 | 1000 |
| Noise | natural | frozen | natural | frozen | natural | frozen | natural | frozen | natural | frozen |
| `random` | 0.3 | 0.0 | 0.7 | 1.3 | 1.9 | 0.3 | 0.2 | 0.1 | 0.1 | 0.1 |
| `cma` | – | 0.1 | – | 5.2 | – | 0.8 | – | 1.3 | – | – |
| `turbo-zero` | 147.3 | 0.5 | 52.6 | 7.8 | 25.9 | 0.7 | 14.9 | 7.9 | 41.2 | 40.0 |
| `optuna` | 3066.2 | 100.6 | 8995.0 | >5hr | 4373.0 | 1142.4 | 3485.9 | 11117.4 | 910.0 | 915.3 |
| `smac` | 13384.7 | 467.7 | >5hr | >5hr | >5hr | >5hr | 15613.9 | >5hr | 1093.9 | 1097.1 |
| `dngo` | >5hr | >5hr | >5hr | >5hr | >5hr | >5hr | >5hr | >5hr | 6742.5 | 7657.1 |
| `ucb` | >5hr | 2210.3 | >5hr | >5hr | >5hr | >5hr | >5hr | >5hr | >5hr | >5hr |
| `turbo-one` | 2128.5 | 52.3 | 17582.5 | 5532.4 | 3048.0 | 173.1 | 3050.0 | 873.8 | 6508.1 | 6596.4 |
| `turbo-enn` | 249.3 | 0.9 | 822.8 | 41.2 | 311.1 | 2.2 | 309.9 | 5.8 | 108.1 | 88.4 |
| speedup | 9 | 58 | 21 | 134 | 10 | 80 | 10 | 150 | 60 | 75 |

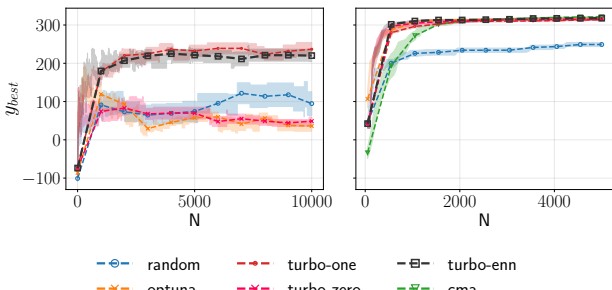

*Figure 5.* BipedalWalker-v3, $D = 16$, using a heuristic controller designed interactively with Cursor (Cursor), GPT-5.2 (OpenAI), and Claude Opus 4.5 (Anthropic). Left: Natural noise. `num_arms = 1`, `num_denoise_passive = 10`. Right: Frozen noise. `num_arms = 50`, `num_denoise_obs = 10`.

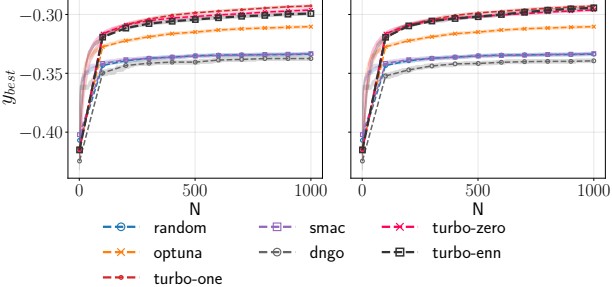

*Figure 7.* LASSO-DNA, $D = 180$, a weighted-LASSO hyperparameter optimization problem from LASSOBench (Šehić et al., 2022). Left: Natural noise. `num_arms = 1`, `num_denoise_passive = 10`. Right: Frozen noise. `num_arms = 1`, `num_denoise_obs = 10`.

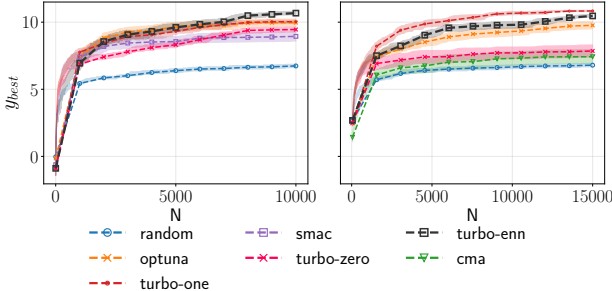

*Figure 6.* Push-v5, $D = 14$, using a heuristic controller presented in (Wang et al., 2017) and (Eriksson et al., 2019b). Left: Natural noise. `num_arms = 1`, `num_denoise_passive = 30`. Right: Frozen noise. `num_arms = 50`, `num_denoise_obs = 50`.

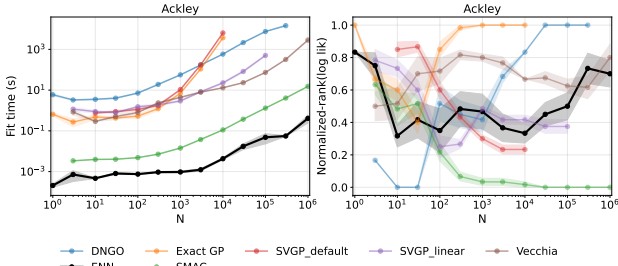

*Figure 8.* Ackley ($D$=10), $N$ i.i.d. fitting points in $[0, 1]^D$ and a test set with $N_{test} = 100$ points (bands: $\pm$ one standard error over 10 replications). Panels report wall-clock fit time on one CPU and/or held-out test-set log-likelihood for the surrogates in Appendix B (same protocol as Figures 16–17).

ing could be improved even further in a surrogate-based BO algorithm. Perhaps ENN could be adapted to an approximate nearest neighbor algorithm (Malkov & Yashunin, 2020) that scales as $O(\ln N)$. We note that the surrogate-free, sampling-only algorithm CMA-ES is constant in $N$, i.e. $O(1)$, and performs very well. The same holds true for our surrogate-free ablation, `turbo-zero`, and we think more attention to the sampling phase of BO may be beneficial.

In this paper we use a constant value of $K$. Future work might explore ways to choose a value of $K$ appropriate to a particular problem or even to a particular observation set. Perhaps $K$ could be tuned along with $s_0$ and $c_e$.

The original TuRBO uses Thompson sampling, an acquisition method that enjoys near-optimal regret for multi-armed bandits (Agrawal & Goyal, 2013) and works very well in Bayesian optimization (Kandasamy et al., 2018b; Sweet, 2024). To create a Thompson sample, the surrogate needs to support joint sampling, but ENN does not yet support this. This could be an interesting avenue for future research.

In GP surrogates, one lengthscale hyperparameter may be assigned to each dimension, a technique called automatic relevance determination (ARD) (Williams & Rasmussen, 1995). Analogous techniques exist for nearest-neighbor models (Li et al., 2015). It may be fruitful to seek a fast, scalable method of weighting dimensions in ENN's distance metric.

As $D$ increases, Euclidean distance (to a query point) will discriminate less between observations, yet RAASP and the trust region may serve to ameliorate this problem. This is an interesting research question.

Scientific and engineering simulations may contain parameter constraints, outcome (metric) constraints, mixed variable types (e.g, continuous, ordinal, categorical), and multiple metrics. These are all aspects of optimization to which ENN should be adapted.

While there is a convergence guarantee, we provide no regret bounds.

## 7. Conclusion

Bayesian optimization with many observations (BOMO) is an increasingly important setting, driven by fast simulations and parallel evaluation, where proposal time becomes significant relative to evaluation time. We proposed TuRBO-ENN, a trust-region Bayesian optimization method that replaces the Gaussian-process surrogate in TuRBO with Epistemic Nearest Neighbors (ENN), a lightweight $K$-nearest-neighbor surrogate that provides both a mean predictor and an uncertainty estimate with linear scaling in $N$ under our implementation. For noisy objectives, TuRBO-ENN fits ENN hyperparameters and selects candidates using UCB.

For deterministic objectives, it admits a fitting-free variant that selects candidates via non-dominated sorting over $(\mu, \sigma)$, further decreasing proposal time. Across the benchmarks we study, including runs with up to $N = 50,000$ observations, TuRBO-ENN achieves solution quality comparable to TuRBO while reducing proposal time by one to two orders of magnitude, with improvements that grow with $N$. Finally, we show in Appendix C that TuRBO-ENN satisfies the Pseudo-Bayesian Optimization axioms, providing a convergence guarantee.

## Acknowledgements

This work was carried out in affiliation with Yeshiva University and supported, in part, by the Katz School of Science and Health Faculty Research Initiative.

## Impact Statement

This paper presents work whose goal is to advance the field of Machine Learning. There are many potential societal consequences of our work, none of which we feel must be specifically highlighted here.

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

# A. Ablations

In this section we show how the performance of TuRBO-ENN varies with the hyperparameters $K$ and $P$ and study the impact of modifying TuRBO's acquisition method. In some examples we find that $K$ and $P$ have little impact on performance, but in others they can have a significant impact.

These studies were conducted on optimizer test functions cataloged in (Surjanovic & Bingham, 2013): `sphere`, a unimodal function, `ackley`, a highly-multimodal function, `booth`, which is plate-shaped (low curvature in multiple directions), and `rosenbrock`, which has a long, narrow valley (low curvature in some directions, high curvature in others).

## A.1. Surrogate Quality vs. $K$ and $P$

*Table 2.* Ackley, D=10, N=1000, P=100

| $K$ | NRMSE | LogLik |
|-----|-------|--------|
| 1 | $1.19 \pm 0.023$ | $-829.43 \pm 31$ |
| 3 | $0.95 \pm 0.011$ | $-728.32 \pm 72$ |
| 10 | $0.86 \pm 0.007$ | $-715.32 \pm 150$ |
| 30 | $0.87 \pm 0.006$ | $-987.02 \pm 341$ |
| 100 | $0.90 \pm 0.009$ | $-1908.58 \pm 666$ |

*Table 3.* Ackley, D=10, N=1000, K=10

| $P$ | NRMSE | LogLik |
|-----|-------|--------|
| 3 | $0.85 \pm 0.008$ | $-1781.36 \pm 536$ |
| 10 | $0.87 \pm 0.007$ | $-1058.67 \pm 364$ |
| 30 | $0.88 \pm 0.009$ | $-1278.77 \pm 238$ |
| 100 | $0.88 \pm 0.008$ | $-1152.75 \pm 321$ |
| 300 | $0.87 \pm 0.012$ | $-1116.69 \pm 249$ |

*Table 4.* Sphere, D=10, N=1000, P=100

| $K$ | NRMSE | LogLik |
|-----|-------|--------|
| 1 | $1.25 \pm 0.005$ | $353.20 \pm 5$ |
| 3 | $1.03 \pm 0.005$ | $535.78 \pm 6$ |
| 10 | $0.94 \pm 0.007$ | $634.54 \pm 8$ |
| 30 | $0.94 \pm 0.005$ | $631.54 \pm 9$ |
| 100 | $0.96 \pm 0.008$ | $622.93 \pm 7$ |

*Table 5.* Sphere, D=10, N=1000, K=10

| $P$ | NRMSE | LogLik |
|-----|-------|--------|
| 3 | $0.93 \pm 0.004$ | $618.74 \pm 13$ |
| 10 | $0.95 \pm 0.004$ | $611.27 \pm 12$ |
| 30 | $0.94 \pm 0.007$ | $613.67 \pm 12$ |
| 100 | $0.94 \pm 0.005$ | $627.95 \pm 8$ |
| 300 | $0.96 \pm 0.008$ | $634.88 \pm 5$ |

Tables 3- 5 show (i) the optimum (minimal NRMSE and maximal LogLik) in $K$ and (ii) decreasing standard error in LogLik with increasing $P$ (see Section 4.1 for discussion of

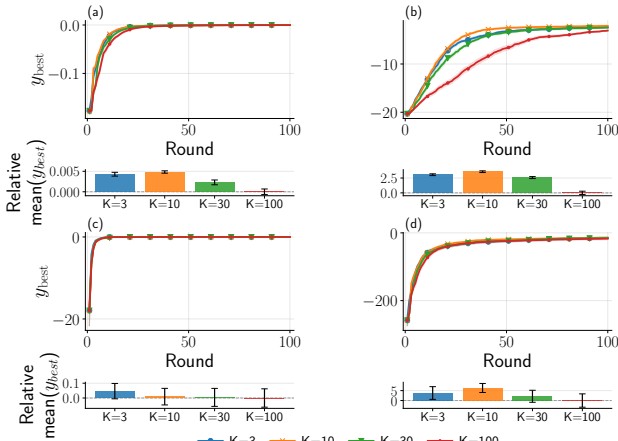

*Figure 9.* $y_{\text{best}}$ vs. $K$ when optimizing the functions (Surjanovic & Bingham, 2013) (a) `sphere`, (b) `ackley`, (c) `booth`, and (d) `rosenbrock`. All functions are 10-dimensional.

$P$).

We define

$$NRMSE(y, \hat{y}) = \frac{\sum_{i=1}^{N}(y_i - \hat{y}_i)^2}{\sum_{i=1}^{N} y_i^2}$$

and

$$\text{LogLik}(y, \hat{y}, v) = \sum_{i=1}^{n}\left[-\frac{1}{2}\log(2\pi v_i) - \frac{(y_i - \hat{y}_i)^2}{2v_i}\right]$$

We fit to synthetic data generated by selecting $N = 1000$ points $x_i \sim \text{Unif}([0, 1]^D)$ and $y_i = f(x_i) + \epsilon_i$ where $\epsilon_i \sim \mathcal{N}(0, 0.1^2)$. The metrics NRMSE and LogLik were computed on a second, independent set of $N = 1000$ points.

## A.2. TuRBO-ENN Sensitivity to $K$ and $P$

Figure 9 shows TuRBO-ENN optimizing noiseless analytical functions, each designed to present different challenges to an optimizer (Surjanovic & Bingham, 2013). We see that the very flat Booth function prefers a small $K$, yet all values of $K$ show good optimization performance. The other functions all prefer a moderate value of $K$ ($K = 10$). Only the highly-multimodal ackley function shows significant optimization performance variation with $K$.

Figure 10 shows the impact of $P$ on TuRBO-ENN's optimization performance on LunarLander-v3 with natural noise and impact of $K$ on the frozen noise problem.

## A.3. Changes to TuRBO's Acquisition Method

Figures 11-15 compare TuRBO-ENN and the original TuRBO (`turbo-one`) to two modifications of TuRBO: (i) `turbo-one-nds`, where Thompson sampling is replaced

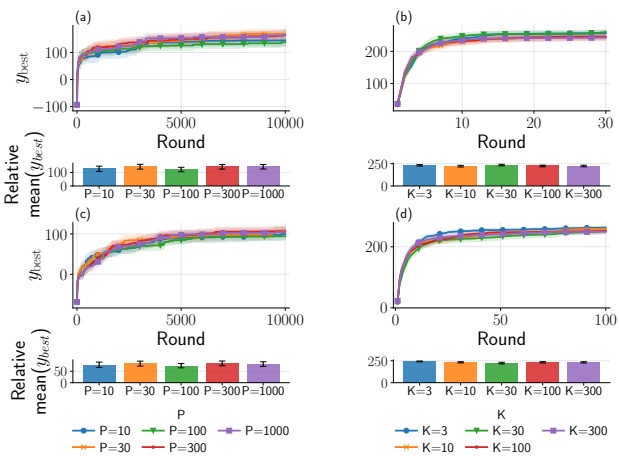

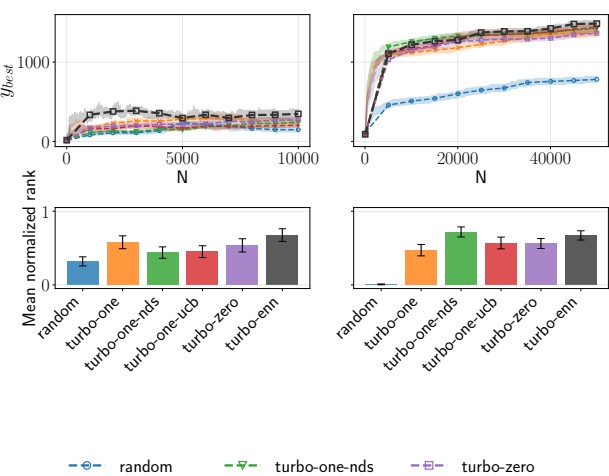

*Figure 10.* $y_{\text{best}}$ vs. $P$ when optimizing (a) LunarLander-v3 with natural noise or (c) BipedalWalker-v3 with natural noise, or $K$ when optimizing (b) LunarLander-v3 with frozen noise or (d) BipedalWalker-v3 with frozen noise. The hyperparameters have little impact in these cases.

*Figure 12.* Hopper-v5, $D = 34$, using a linear controller, similar to (Mania et al., 2018). Left: Natural noise. num_arms = 1, num_denoise_passive = 10. Right: Frozen noise. num_arms = 50, num_denoise_obs = 10.

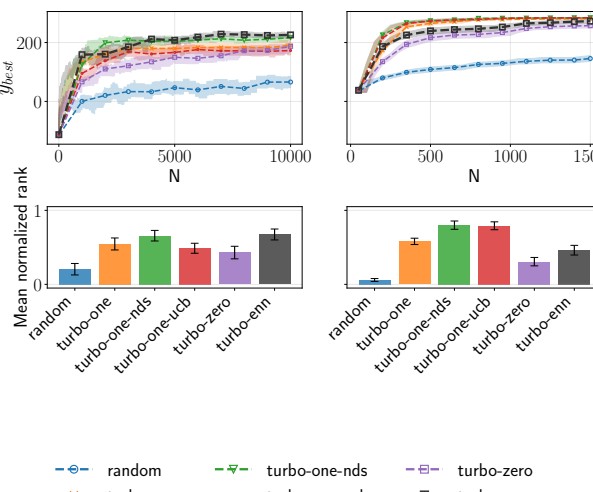

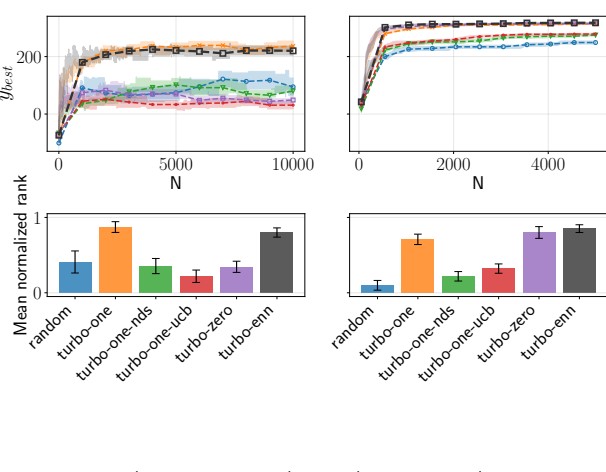

*Figure 11.* LunarLander-v3, $D = 12$, using the controller presented in (Eriksson et al., 2019b). Left: Natural noise. num_arms = 1, num_denoise_passive = 30. Right: Frozen noise. num_arms = 50, num_denoise_obs = 50.

*Figure 13.* BipedalWalker-v3, $D = 16$, using a heuristic controller designed interactively with Cursor (Cursor), GPT-5.2 (OpenAI), and Claude Opus 4.5 (Anthropic). Left: Natural noise. num_arms = 1, num_denoise_passive = 10. Right: Frozen noise. num_arms = 50, num_denoise_obs = 10.

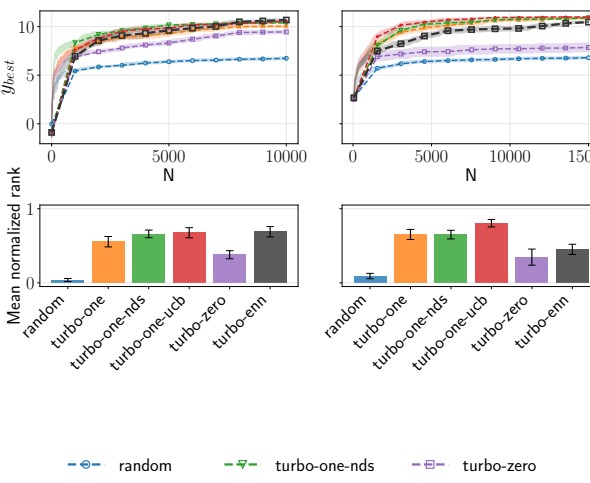

*Figure 14.* Push-v5, $D = 14$, using a heuristic controller presented in (Wang et al., 2017) and (Eriksson et al., 2019b). Left: Natural noise. num_arms = 1, num_denoise_passive = 30. Right: Frozen noise. num_arms = 50, num_denoise_obs = 50.

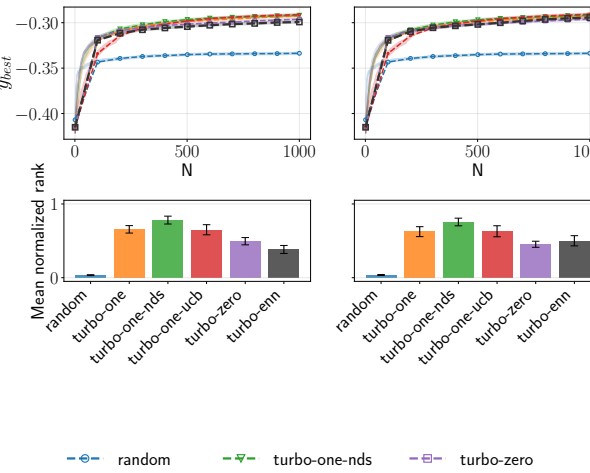

*Figure 15.* LASSO-DNA, $D = 180$, a weighted-LASSO hyperparameter optimization problem from LASSOBench (Šehić et al., 2022). Left: Natural noise. num_arms = 1, num_denoise_passive = 10. Right: Frozen noise. num_arms = 1, num_denoise_obs = 10.

by non-dominated sorting, and (ii) turbo-one-ucb, where Thompson sampling is replaced by UCB. Mean normalized rank is computed by ranking the optimization methods at each round by each method's best observed objective $y_{best}$ at that round, range-normalizing to $[0, 1]$, then averaging over all rounds. The ablations turbo-one-nds and turbo-one-ucb are neither consistently better nor worse than turbo-one.

## B. Fitting Time and Quality

Here we compare various surrogate models on pure regression tasks. Figure 16 shows that ENN takes less time to fit to samples of sizes N=1 through N=10,000 than the other surrogates tested, while Figure 17 shows that ENN's test-set predictions consistently rank near the median (or better) across surrogates. Note that for some surrogates no code path defines a fit when N=1, so these points are omitted from the plots. Also, some surrogates took longer than our 5-hour time limit (using one CPU for each run), so we also omitted those (large-N) points. An interesting point of comparison is (Wang et al., 2019), a 2019 paper that reports, "an exact GP can be trained on over a million points ... in less than 2 hours" by leveraging multi-GPU parallelization. Our plots omit $N = 10^6$ for the exact GP surrogate because it took longer than our 5-hour time limit on a single CPU, but note that ENN on a single CPU for $N = 10^6$ points took only $\sim 0.3$ seconds.

The surrogates tested were discussed in Section 3 and are labeled

- Exact GP: Exact Gaussian Process

- SVGP_default: Sparse Gaussian Process with $M = 0.25N$ inducing points (Titsias, 2009)

- SVGP_linear: Sparse Gaussian Process with $M = \min(1000, N)$ inducing points (Titsias, 2009)

- DNGO: Neural Network surrogate (DNGO, (Snoek et al., 2015))

- SMAC: Random Forest surrogate (SMAC, (Hutter et al., 2011))

- Vecchia: Vecchia Gaussian Process (Jimenez & Katzfuss, 2023)

- ENN: Epistemic Nearest Neighbors

## C. ENN as a Surrogate Model for Bayesian Optimization

This appendix develops the theory of ENN as a surrogate model for Bayesian optimization. Under exact observations, we verify local consistency, sequential no-empty-ball,

and an auxiliary improvement property for the ENN mean–uncertainty pair. Under noisy observations, we interpret ENN as a precision-weighted $k$NN estimator and derive a bias–variance bound and local rate. We then show that these conclusions transfer to fitted variance-scale parameters under a short stability condition.

### C.1. Surrogate model properties

Let $\mathcal{X} \subset \mathbb{R}^D$ be compact and let $f : \mathcal{X} \to \mathbb{R}$ be continuous. We observe $y_i = f(x_i)$, write $\mathcal{X}_n = \{x_1, \ldots, x_n\}$ for the set of sampled inputs, and set $K_n = \min\{K, n\}$. Thus $K_n$ is the number of observed neighbors used at round $n$; once $n \geq K$, it equals the fixed neighbor cap $K$. For each query point $x \in \mathcal{X}$, let

$$z_{n,1}(x), \ldots, z_{n,K_n}(x)$$

denote the $K_n$ nearest observed inputs to $x$ drawn from the observation list $x_1, \ldots, x_n$, ordered so that

$$\|x - z_{n,1}(x)\|_2 \leq \cdots \leq \|x - z_{n,K_n}(x)\|_2,$$

with deterministic tie-breaking. Thus $z_{n,i}(x)$ is the $i$th nearest sampled input to the query point $x$. We write $r_{n,i}(x)$ for the corresponding query-to-neighbor distance, and $d(x, \mathcal{X}_n)$ for the distance from $x$ to the sampled design:

$$r_{n,i}(x) := \|x - z_{n,i}(x)\|_2, \qquad i = 1, \ldots, K_n,$$
$$d(x, \mathcal{X}_n) := \min_{z \in \mathcal{X}_n} \|x - z\|_2 = r_{n,1}(x).$$

For $x \notin \mathcal{X}_n$, ENN assigns inverse-square weights to these $K_n$ neighbors, then forms a mean-like prediction $\mu_n(x)$ and an uncertainty-like scale $\sigma_n(x)$:

$$w_{n,i}(x) := \frac{r_{n,i}(x)^{-2}}{\sum_{j=1}^{K_n} r_{n,j}(x)^{-2}},$$

$$\mu_n(x) := \sum_{i=1}^{K_n} w_{n,i}(x) f(z_{n,i}(x)),$$

$$\sigma_n^2(x) := \left(\sum_{i=1}^{K_n} r_{n,i}(x)^{-2}\right)^{-1}. \tag{4}$$

At sampled points, we use the interpolating convention $\mu_n(x) = f(x)$ and $\sigma_n(x) = 0$, so the surrogate mean interpolates the observed value and the uncertainty vanishes at sampled inputs.

**Theorem 1** (ENN surrogate properties under exact observations)**.** *For the ENN construction in equation 4, the mean $\mu_n$ and uncertainty $\sigma_n$ have the following properties.*

*(i)* Deterministic approximation. *If $\omega_f(r) = \sup\{|f(u) - f(v)| : \|u - v\|_2 \leq r\}$, then for $x \notin$*

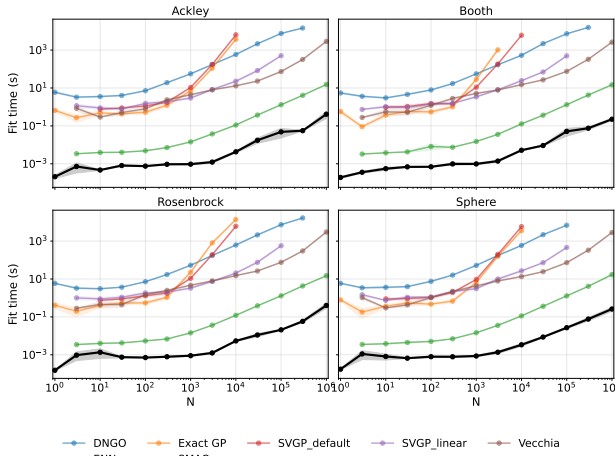

*Figure 16.* ENN takes less time – and often significantly less time (note the log-scaled y axis) – to fit than the other surrogates.

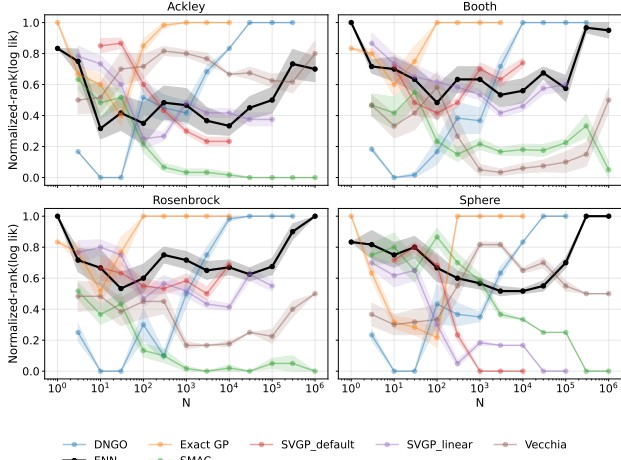

*Figure 17.* ENN's test-set log-likelihood (see Section A.1 for definition) ranks near the median (or better) across surrogates.

$\mathcal{X}_n$,

$$|\mu_n(x) - f(x)| \leq \sum_{i=1}^{K_n} w_{n,i}(x)\omega_f(r_{n,i}(x)) \tag{5}$$
$$\leq \omega_f(r_{n,K_n}(x)).$$

*If f is L-Lipschitz, then*

$$|\mu_n(x) - f(x)| \leq L \sum_{i=1}^{K_n} w_{n,i}(x)r_{n,i}(x). \tag{6}$$

(ii) Spacing equivalence. *For every $x \notin \mathcal{X}_n$,*

$$\frac{d(x, \mathcal{X}_n)}{\sqrt{K_n}} \leq \sigma_n(x) \leq d(x, \mathcal{X}_n). \tag{7}$$

(iii) Local consistency. *The mean sequence $(\mu_n)$ is locally consistent: for every $x_0 \in \mathcal{X}$ and every sequence $(\xi_m)_{m \geq 1}$ with $\xi_m \in \mathcal{X}_{n_m}$ and $\xi_m \to x_0$, one has $\mu_{n_m}(x_0) \to f(x_0)$.*

(iv) Sequential no-empty-ball. *The uncertainty sequence $(\sigma_n)$ satisfies sequential no-empty-ball: if $B(x,r) \cap \mathcal{X}_n = \emptyset$, then $\sigma_n(x) \geq r/\sqrt{K}$. Moreover, if $(\xi_m)_{m \geq 1}$ satisfies $\xi_m \in \mathcal{X}_{n_m}$ and $\xi_m \to x_0$, then $\sigma_{n_m}(x_0) \to 0$.*

*Proof.* Fix $n$ and $x \notin \mathcal{X}_n$. Write

$$z_i := z_{n,i}(x), \qquad r_i := r_{n,i}(x),$$
$$w_i := w_{n,i}(x), \qquad d := d(x, \mathcal{X}_n).$$

We begin with

$$\mu_n(x) - f(x) = \sum_{i=1}^{K_n} w_i\{f(z_i) - f(x)\},$$

so

$$|\mu_n(x) - f(x)| \leq \sum_{i=1}^{K_n} w_i \left|f(z_i) - f(x)\right|$$
$$\leq \sum_{i=1}^{K_n} w_i \, \omega_f(r_i) \tag{8}$$
$$\leq \omega_f(r_{K_n}).$$

Here we used that the weights are nonnegative and sum to one, that $|f(z_i) - f(x)| \leq \omega_f(r_i)$, and that $\omega_f$ is non-decreasing. If $f$ is L-Lipschitz, then $\omega_f(r) \leq Lr$, and therefore

$$|\mu_n(x) - f(x)| \leq L \sum_{i=1}^{K_n} w_i r_i. \tag{9}$$

For the spacing bound,

$$r_1 = d,$$
$$r_i \geq d, \qquad 1 \leq i \leq K_n.$$

Hence

$$\frac{1}{d^2} \leq \sum_{i=1}^{K_n} \frac{1}{r_i^2}$$
$$\leq \frac{K_n}{d^2},$$

and all three quantities are strictly positive because $x \notin \mathcal{X}_n$. Therefore

$$\frac{d^2}{K_n} \leq \left(\sum_{i=1}^{K_n} \frac{1}{r_i^2}\right)^{-1} \leq d^2. \tag{10}$$

By the definition of $\sigma_n(x)^2$, equation 10 becomes

$$\frac{d^2}{K_n} \leq \sigma_n(x)^2 \leq d^2. \tag{11}$$

Since $\sigma_n(x) \geq 0$ and $d \geq 0$, equation 11 yields

$$\frac{d}{\sqrt{K_n}} \leq \sigma_n(x) \leq d. \tag{12}$$

Fix $x_0 \in \mathcal{X}$ and a sequence $(\xi_m)_{m \geq 1}$ with $\xi_m \in \mathcal{X}_{n_m}$ and $\xi_m \to x_0$. Write

$$r_{m,i} := r_{n_m,i}(x_0), \qquad w_{m,i} := w_{n_m,i}(x_0),$$
$$d_m := d(x_0, \mathcal{X}_{n_m}).$$

We prove local consistency. If $x_0$ has already been sampled, interpolation makes the claim immediate. Otherwise $d_m \leq \|x_0 - \xi_m\|_2 \to 0$. Fix $\eta > 0$ and choose $\delta > 0$ such that $\|x - x_0\|_2 < \delta$ implies $|f(x) - f(x_0)| < \eta$. Let $A_m = \{i : r_{m,i} < \delta\}$ and $B_m = \{1, \ldots, K_{n_m}\} \setminus A_m$. With $M = \sup_{\mathcal{X}} |f|$,

$$|\mu_{n_m}(x_0) - f(x_0)| \leq \eta + 2M \sum_{i \in B_m} w_{m,i}. \tag{13}$$

For $i \in B_m$, $r_{m,i}^{-2} \leq \delta^{-2}$, while the denominator in the ENN weight is at least $d_m^{-2}$. Therefore

$$\sum_{i \in B_m} w_{m,i} \leq K\delta^{-2}d_m^2 \to 0. \tag{14}$$

Combining equation 13 and equation 14, taking the limsup, and then letting $\eta \downarrow 0$ proves (iii).

For sequential no-empty-ball, if $B(x,r) \cap \mathcal{X}_n = \emptyset$, then every ENN neighbor has distance at least $r$, so equation 7 gives $\sigma_n(x) \geq r/\sqrt{K}$. For the same sequence $(\xi_m)_{m \geq 1}$, either $x_0$ is eventually sampled, in which case $\sigma_{n_m}(x_0) = 0$, or $d_m \leq \|x_0 - \xi_m\|_2 \to 0$, and equation 7 gives $\sigma_{n_m}(x_0) \to 0$. $\square$

## C.2. Acquisition Geometry

At each candidate $x$, ENN supplies the pseudo-posterior pair $(\mu_n(x), \sigma_n(x))$: a mean-like prediction and an uncertainty-like scale. BO routines may use this pair through set-valued rules such as nondominated sorting, or through scalar rules such as UCB. For the Pseudo-Bayesian Optimization (PBO) scalar acquisition formalism (Chen & Lam, 2025), we introduce the auxiliary scalarization

$$m_n := \inf_{z \in \mathcal{X}} \mu_n(z),$$
$$\widetilde{\alpha}_n^\lambda(x) := \lambda(\mu_n(x) - m_n)^+ + (1 - \lambda)\sigma_n(x),$$

where $\lambda \in (0, 1)$ and $a^+$ denotes the positive part of $a$. Since $m_n = \inf_{z \in \mathcal{X}} \mu_n(z)$, we have $\mu_n(x) - m_n \geq 0$ for every $x$, and hence

$$\widetilde{\alpha}_n^\lambda(x) = \lambda\mu_n(x) + (1 - \lambda)\sigma_n(x) - \lambda m_n. \tag{15}$$

Therefore, over any fixed candidate set, maximizing $\widetilde{\alpha}_n^\lambda$ is equivalent to maximizing a positive linear scalarization of the pseudo-posterior pair:

$$x \mapsto \lambda\mu_n(x) + (1 - \lambda)\sigma_n(x). \tag{16}$$

The same coordinate order also underlies UCB and NDS: UCB chooses one positive scalarization direction, whereas NDS retains the entire first Pareto front. In particular, every maximizer of a positive scalarization is nondominated with respect to $(\mu_n, \sigma_n)$.

**Proposition 1** (Improvement property of the scalarized ENN acquisition). *Fix $\lambda \in (0, 1)$ and set*

$$A_\lambda(g, u) := \lambda g^+ + (1 - \lambda)u, \qquad g \in \mathbb{R}, \ u \geq 0.$$

*Then $A_\lambda$ has the Chen–Lam improvement property (Chen & Lam, 2025): for any real sequence $(g_n)$ and nonnegative sequence $(u_n)$,*

$$\liminf_n A_\lambda(g_n, u_n) > 0$$

*whenever $\liminf_n g_n > -\infty$ and $\liminf_n u_n > 0$, while $A_\lambda(g_n, u_n) \to 0$ whenever $\limsup_n g_n \leq 0$ and $u_n \to 0$. Consequently, $x \mapsto \widetilde{\alpha}_n^\lambda(x)$ satisfies the improvement condition. Moreover, every maximizer of a positive linear scalarization of $(\mu_n, \sigma_n)$ over a finite candidate set is nondominated with respect to the same two coordinates.*

*Proof.* If $\liminf u_n > 0$ and $\liminf g_n > -\infty$, then $\liminf A_\lambda(g_n, u_n) > 0$. If $\limsup g_n \leq 0$ and $u_n \to 0$, then $g_n^+ \to 0$ and $A_\lambda(g_n, u_n) \to 0$. This proves the improvement property. In particular, for any sequence $(x_n)$, if we set

$$g_n := \mu_n(x_n) - m_n,$$
$$u_n := \sigma_n(x_n),$$

then

$$\liminf_n \widetilde{\alpha}_n^\lambda(x_n) > 0$$

whenever

$$\liminf_n (\mu_n(x_n) - m_n) > -\infty,$$
$$\liminf_n \sigma_n(x_n) > 0,$$

while

$$\widetilde{\alpha}_n^\lambda(x_n) \to 0$$

whenever

$$\limsup_n (\mu_n(x_n) - m_n) \leq 0,$$
$$\sigma_n(x_n) \to 0.$$

If a candidate is dominated in both $\mu_n$ and $\sigma_n$, with one inequality strict, every positive linear scalarization assigns it a strictly smaller value than the dominating candidate. $\square$

Together, Theorem 1 and Proposition 1 verify the surrogate-side local-consistency, sequential no-empty-ball, and improvement conditions of the PBO framework (Chen & Lam, 2025) for the ENN mean–uncertainty pair and the auxiliary scalarization.

## C.3. Weighted $k$NN Analysis

We retain the geometric notation from the exact-observation setting and use tildes to distinguish the noisy weighted-$k$NN quantities from their exact-observation ENN counterparts.

Fix a query point $x \in \mathcal{X}$. Suppose $f$ is $L$-Lipschitz and the observations satisfy

$$y_i = f(x_i) + \varepsilon_i, \qquad i = 1, 2, \ldots.$$

Let $\mathcal{G}_n = \sigma((x_i, s_i)_{i=1}^n)$. Conditional on $\mathcal{G}_n$, the noises are independent and satisfy

$$\mathbb{E}[\varepsilon_i \mid \mathcal{G}_n] = 0,$$
$$\mathrm{Var}(\varepsilon_i \mid \mathcal{G}_n) \leq \bar{\sigma}^2, \qquad i \leq n.$$

Let $1 \leq k_n \leq n$, and let

$$z_{n,1}, \ldots, z_{n,n}$$

denote the observation list $x_1, \ldots, x_n$ reordered so that

$$\|x - z_{n,1}\|_2 \leq \cdots \leq \|x - z_{n,n}\|_2.$$

For $i \leq k_n$, let $y_{n,i}$, $\varepsilon_{n,i}$, and $s_{n,i}$ denote the response, noise, and scale attached to $z_{n,i}$, and define

$$r_{n,i} := \|x - z_{n,i}\|_2.$$

Then $y_{n,i} = f(z_{n,i}) + \varepsilon_{n,i}$. Let $\rho_n(x) = r_{n,k_n}$. For fixed $s_0, c_e > 0$, define

$$v_{n,i} := s_0^2 + s_{n,i}^2 + c_e r_{n,i}^2,$$

$$\widetilde{w}_{n,i} := \frac{v_{n,i}^{-1}}{\sum_{j=1}^{k_n} v_{n,j}^{-1}},$$

$$\widetilde{\mu}_n := \sum_{i=1}^{k_n} \widetilde{w}_{n,i} y_{n,i}.$$

**Theorem 2** (Weighted-$k$NN bias–variance bound and local rate). *Under the noisy design conditions above, the ENN weighted average satisfies the conditional bias–variance bound*

$$\mathbb{E}\big[(\widetilde{\mu}_n - f(x))^2 \mid \mathcal{G}_n\big] \leq L^2 \rho_n(x)^2 \tag{17}$$
$$+ \bar{\sigma}^2 \sum_{i=1}^{k_n} \widetilde{w}_{n,i}^2.$$

*Consequently:*

*(i) If*

$$\mathbb{E}[\rho_n(x)^2] \to 0,$$

$$\mathbb{E}\left[\sum_{i=1}^{k_n} \widetilde{w}_{n,i}^2\right] \to 0,$$

*then $\widetilde{\mu}_n \to f(x)$ in $L^2$ and hence in probability.*

*(ii) Write $u_n \lesssim v_n$ when $u_n \leq C v_n$ eventually for a constant $C < \infty$ independent of $n$. If*

$$\mathbb{E}[\rho_n(x)^2] \lesssim a_n(x)^2,$$

$$\mathbb{E}\left[\sum_{i=1}^{k_n} \widetilde{w}_{n,i}^2\right] \lesssim \frac{1}{k_n},$$

*then*

$$\mathbb{E}[(\widetilde{\mu}_n - f(x))^2] \lesssim a_n(x)^2 + \frac{1}{k_n}.$$

*In particular, if*

$$a_n(x) \asymp \left(\frac{k_n}{n}\right)^{1/q} \quad \text{and} \quad k_n \asymp n^{\frac{2}{q+2}},$$

*then*

$$\mathbb{E}[(\widetilde{\mu}_n - f(x))^2] \lesssim n^{-\frac{2}{q+2}}.$$

*Proof.* For brevity, write

$$z_i := z_{n,i}, \qquad r_i := r_{n,i},$$
$$\widetilde{w}_i := \widetilde{w}_{n,i}, \qquad \varepsilon_i := \varepsilon_{n,i},$$
$$\rho := \rho_n(x).$$

Write

$$B_n := \sum_{i=1}^{k_n} \widetilde{w}_i \{f(z_i) - f(x)\},$$

$$N_n := \sum_{i=1}^{k_n} \widetilde{w}_i \varepsilon_i.$$

Then $\widetilde{\mu}_n - f(x) = B_n + N_n$. The reordered noises remain conditionally independent and centered given $\mathcal{G}_n$, while the weights are $\mathcal{G}_n$-measurable. Hence

$$\mathbb{E}[N_n \mid \mathcal{G}_n] = 0,$$

$$\mathbb{E}[N_n^2 \mid \mathcal{G}_n] \leq \bar{\sigma}^2 \sum_{i=1}^{k_n} \widetilde{w}_i^2.$$

Since the weights sum to one and $r_i \leq \rho$, Lipschitz continuity gives $|B_n| \leq L\rho$. The cross term vanishes conditionally, proving the conditional bias–variance bound equation 17. Taking expectations in equation 17 yields

$$\mathbb{E}[(\widetilde{\mu}_n - f(x))^2] \leq L^2 \mathbb{E}[\rho_n(x)^2] + \bar{\sigma}^2 \mathbb{E}\left[\sum_{i=1}^{k_n} \widetilde{w}_i^2\right]. \tag{18}$$

This gives both the consistency claim in (i) and the rate bound in (ii). In particular, if $\mathbb{E}[(\widetilde{\mu}_n - f(x))^2] \to 0$, then applying Chebyshev's inequality yields, for every $\varepsilon > 0$,

$$\mathbb{P}\{|\widetilde{\mu}_n - f(x)| > \varepsilon\} \leq \frac{\mathbb{E}[(\widetilde{\mu}_n - f(x))^2]}{\varepsilon^2} \to 0,$$

so $\widetilde{\mu}_n \to f(x)$ in probability. $\square$

The two terms in Theorem 2 are the usual weighted-$k$NN quantities: the radius $\rho_n(x)$ controls bias, and $\sum_i \widetilde{w}_{n,i}^2$ is the effective inverse sample size for noise. Local small-ball lower bounds such as $\mathbb{P}\{X \in B(x,r)\} \gtrsim r^q$ verify the radius scale under standard random-design conditions.

### C.4. Fitted Variance-Scale Parameters

The implementation fits $\theta = (s_0, c_e)$. As in Section 4, for each $n \geq 2$ let $\ell_{-i}(\theta)$ denote the leave-one-out Gaussian log score formed from the remaining $n - 1$ observations, so the underlying ENN predictor uses $\min\{K, n-1\}$ neighbors. Define the full average LOO log score by

$$\bar{\ell}_n(\theta) := \frac{1}{n} \sum_{i=1}^{n} \ell_{-i}(\theta).$$

For analysis, use the equivalent variance-scale coordinates $\phi = (\alpha, \gamma) = (s_0^2, c_e)$ on a compact set $\Phi = [\underline{\alpha}, \bar{\alpha}] \times [\underline{\gamma}, \bar{\gamma}]$, where $0 < \underline{\alpha} \leq \bar{\alpha} < \infty$ and $0 \leq \underline{\gamma} \leq \bar{\gamma} < \infty$. Write

$$\theta(\phi) := (\alpha^{1/2}, \gamma), \qquad L_n(\phi) := \bar{\ell}_n(\theta(\phi)),$$

Let $Q_n(\phi) = \mathbb{E}[L_n(\phi) \mid (x_i, s_i)_{i=1}^n]$. Assume $L_n$ and $Q_n$ are continuous on $\Phi$, so their argmax sets are nonempty. Let

$$\hat{\phi}_n \in \arg\max_{\phi \in \Phi} L_n(\phi),$$

$$\phi_n^\star \in \arg\max_{\phi \in \Phi} Q_n(\phi).$$

**Theorem 3** (Fitted variance-scale transfer). *Assume*

$$\sup_{\phi \in \Phi} \left| L_n(\phi) - Q_n(\phi) \right| \xrightarrow{\mathbb{P}} 0,$$

*and that the conditional criterion has a separated target maximizer: for every $\delta > 0$, there exists $\eta(\delta) > 0$ such that*

$$\mathbb{P}\left\{ Q_n(\phi_n^\star) - \sup_{\|\phi - \phi_n^\star\|_1 \geq \delta} Q_n(\phi) \geq \eta(\delta) \right\} \to 1.$$

*Then $\|\hat{\phi}_n - \phi_n^\star\|_1 \to 0$ in probability.*

*For a fixed query point $x$, define*

$$\widetilde{w}_{n,i}(\phi) := \frac{(\alpha + s_{n,i}^2 + \gamma r_{n,i}^2)^{-1}}{\sum_{j=1}^{k_n} (\alpha + s_{n,j}^2 + \gamma r_{n,j}^2)^{-1}},$$

$$\widetilde{\mu}_n(x; \phi) := \sum_{i=1}^{k_n} \widetilde{w}_{n,i}(\phi) y_{n,i}.$$

*Set*

$$s_n^\star(x) := \sup_{i \leq k_n} s_{n,i},$$

$$Y_n^\star(x) := \sup_{i \leq k_n} |y_{n,i}|,$$

$$V_n(x) := \bar{\alpha} + \left(s_n^\star(x)\right)^2 + \bar{\gamma} \rho_n(x)^2,$$

$$\Lambda_n(x) := \frac{2 V_n(x)(1 + \rho_n(x)^2)}{\underline{\alpha}^2},$$

$$\Gamma_n(x) := Y_n^\star(x) \Lambda_n(x),$$

*Then, for every $\phi, \phi' \in \Phi$,*

$$\sum_{i=1}^{k_n} \left| \widetilde{w}_{n,i}(\phi) - \widetilde{w}_{n,i}(\phi') \right| \leq \Lambda_n(x) \|\phi - \phi'\|_1, \qquad (19)$$

*and therefore*

$$\left| \widetilde{\mu}_n(x; \phi) - \widetilde{\mu}_n(x; \phi') \right| \leq \Gamma_n(x) \|\phi - \phi'\|_1. \qquad (20)$$

*Consequently, if*

$$\widetilde{\mu}_n(x; \phi_n^\star) \to f(x) \quad \text{in probability},$$
$$\Gamma_n(x) = O_{\mathbb{P}}(1),$$

*then $\widetilde{\mu}_n(x; \hat{\phi}_n) \to f(x)$ in probability. If instead*

$$\widetilde{\mu}_n(x; \phi_n^\star) \to f(x) \quad \text{in } L^2,$$

$$\mathbb{E}\left[ \Gamma_n(x)^2 \|\hat{\phi}_n - \phi_n^\star\|_1^2 \right] \to 0,$$

*then $\widetilde{\mu}_n(x; \hat{\phi}_n) \to f(x)$ in $L^2$.*

*Proof.* We fix $\delta > 0$ and define

$$\Delta_n := \sup_{\phi \in \Phi} \left| L_n(\phi) - Q_n(\phi) \right|,$$

$$E_n(\delta) := \left\{ Q_n(\phi_n^\star) - \sup_{\|\phi - \phi_n^\star\|_1 \geq \delta} Q_n(\phi) \geq \eta(\delta) \right\},$$

$$F_n(\delta) := \{\Delta_n \leq \eta(\delta)/3\}.$$

By assumption, $\mathbb{P}\{E_n(\delta)\} \to 1$ and $\mathbb{P}\{F_n(\delta)\} \to 1$. On the event $E_n(\delta) \cap F_n(\delta)$, every $\phi$ satisfying $\|\phi - \phi_n^\star\|_1 \geq \delta$ obeys

$$\begin{aligned}
L_n(\phi) &\leq Q_n(\phi) + \Delta_n \\
&\leq Q_n(\phi_n^\star) - \eta(\delta) + \Delta_n \\
&\leq Q_n(\phi_n^\star) - 2\eta(\delta)/3 \\
&\leq L_n(\phi_n^\star) - \eta(\delta)/3.
\end{aligned}$$

Hence no maximizer of $L_n$ lies outside the $\delta$-ball around $\phi_n^\star$ on $E_n(\delta) \cap F_n(\delta)$. Therefore

$$\mathbb{P}\{\|\hat{\phi}_n - \phi_n^\star\|_1 \geq \delta\} \leq \mathbb{P}\{E_n(\delta)^c\} + \mathbb{P}\{F_n(\delta)^c\} \to 0,$$

which proves $\|\hat{\phi}_n - \phi_n^\star\|_1 \to 0$ in probability.

We now prove the stability estimate. For brevity, write

$$\begin{aligned}
s_i &:= s_{n,i}, & r_i &:= r_{n,i}, \\
y_i &:= y_{n,i}, & \rho &:= \rho_n(x), \\
V &:= V_n(x), & \Lambda &:= \Lambda_n(x), \\
\Gamma &:= \Gamma_n(x),
\end{aligned}$$

and define

$$u_i(\phi) := (\alpha + s_i^2 + \gamma r_i^2)^{-1}, \qquad U(\phi) := \sum_{i=1}^{k_n} u_i(\phi).$$

For $\phi, \phi' \in \Phi$, we first prove equation 19. We claim that

$$|u_i(\phi) - u_i(\phi')| \leq \frac{1 + \rho^2}{\underline{\alpha}^2} \|\phi - \phi'\|_1.$$

Indeed, writing $\phi = (\alpha, \gamma)$ and $\phi' = (\alpha', \gamma')$, and setting

$$D_i(\phi) := \alpha + s_i^2 + \gamma r_i^2,$$

we have

$$\begin{aligned}
|u_i(\phi) - u_i(\phi')| &= \frac{|D_i(\phi') - D_i(\phi)|}{D_i(\phi) D_i(\phi')} \\
&\leq \frac{|\alpha - \alpha'|}{\underline{\alpha}^2} + \frac{r_i^2 |\gamma - \gamma'|}{\underline{\alpha}^2} \\
&\leq \frac{1 + \rho^2}{\underline{\alpha}^2} \|\phi - \phi'\|_1.
\end{aligned}$$

Because $\phi, \phi' \in \Phi$, $s_i^2 \leq \left(s_n^\star(x)\right)^2$, and $r_i^2 \leq \rho^2$, we also have

$$u_i(\phi),\ u_i(\phi') \geq \frac{1}{V},$$

$$U(\phi),\ U(\phi') \geq \frac{k_n}{V}.$$

We now write $\Delta u_i := u_i(\phi) - u_i(\phi')$. Then

$$\sum_{i=1}^{k_n} |\Delta u_i| \leq \frac{k_n(1+\rho^2)}{\underline{\alpha}^2} \|\phi - \phi'\|_1,$$

$$|U(\phi) - U(\phi')| \leq \sum_{i=1}^{k_n} |\Delta u_i|.$$

Write $\widetilde{w}_i(\phi) := \widetilde{w}_{n,i}(\phi)$. Since $\widetilde{w}_i(\phi) = u_i(\phi)/U(\phi)$, we obtain

$$\left|\widetilde{w}_i(\phi) - \widetilde{w}_i(\phi')\right| \leq \frac{|\Delta u_i|}{U(\phi)} + \frac{u_i(\phi')|U(\phi) - U(\phi')|}{U(\phi)U(\phi')}.$$

We now write

$$\Delta w_i := \widetilde{w}_i(\phi) - \widetilde{w}_i(\phi').$$

Summing over $i \leq k_n$ and using $\sum_{i=1}^{k_n} u_i(\phi') = U(\phi')$, we get

$$\sum_{i=1}^{k_n} |\Delta w_i| \leq \frac{\sum_{i=1}^{k_n} |\Delta u_i|}{U(\phi)} + \frac{|U(\phi) - U(\phi')|}{U(\phi)}$$

$$\leq \frac{2\sum_{i=1}^{k_n} |\Delta u_i|}{U(\phi)}$$

$$\leq \frac{2V}{k_n} \sum_{i=1}^{k_n} |\Delta u_i|$$

$$\leq \Lambda \|\phi - \phi'\|_1.$$

Combining the previous display with the definition of $\Gamma$ gives

$$|\widetilde{\mu}_n(x;\phi) - \widetilde{\mu}_n(x;\phi')| \leq \sum_{i=1}^{k_n} |y_i|\,|\Delta w_i|$$

$$\leq Y_n^\star(x) \sum_{i=1}^{k_n} |\Delta w_i| \qquad (21)$$

$$\leq \Gamma \|\phi - \phi'\|_1.$$

This is exactly equation 20.

We now prove the probabilistic transfer statement. We write

$$\widehat{m}_n := \widetilde{\mu}_n(x;\hat{\phi}_n),$$
$$m_n^\star := \widetilde{\mu}_n(x;\phi_n^\star),$$
$$\Delta\phi_n := \|\hat{\phi}_n - \phi_n^\star\|_1.$$

Then equation 21 gives

$$|\widehat{m}_n - m_n^\star| \leq \Gamma\Delta\phi_n. \qquad (22)$$

We claim that $\Gamma\Delta\phi_n \to 0$ in probability. To prove this, we fix $\varepsilon > 0$ and $\eta > 0$. Since $\Gamma = \Gamma_n(x) = O_{\mathbb{P}}(1)$, there exists $M > 0$ such that

$$\limsup_{n\to\infty} \mathbb{P}\{\Gamma > M\} \leq \eta.$$

For this choice of $M$, we have

$$\mathbb{P}\{\Gamma\Delta\phi_n > \varepsilon\} \leq \mathbb{P}\{\Gamma > M\}$$
$$+ \mathbb{P}\{\Delta\phi_n > \varepsilon/M\}.$$

Since $\Delta\phi_n \to 0$ in probability, the second term converges to zero, and hence

$$\limsup_{n\to\infty} \mathbb{P}\{\Gamma\Delta\phi_n > \varepsilon\} \leq \eta.$$

Here $o_{\mathbb{P}}(1)$ denotes convergence to zero in probability. Because $\eta > 0$ is arbitrary, we conclude that $\Gamma\Delta\phi_n = o_{\mathbb{P}}(1)$. Since $|\widehat{m}_n - m_n^\star| \leq \Gamma\Delta\phi_n$,

$$\widehat{m}_n - m_n^\star = o_{\mathbb{P}}(1).$$

Under the oracle probability assumption, $m_n^\star - f(x) = o_{\mathbb{P}}(1)$. Therefore

$$\widehat{m}_n - f(x) = (\widehat{m}_n - m_n^\star) + (m_n^\star - f(x)) = o_{\mathbb{P}}(1),$$

which proves $\widetilde{\mu}_n(x;\hat{\phi}_n) \to f(x)$ in probability.

For the $L^2$ claim, write $\|Z\|_{L^2} := (\mathbb{E}[Z^2])^{1/2}$. The triangle inequality gives

$$\|\widehat{m}_n - f(x)\|_{L^2} \leq \|\widehat{m}_n - m_n^\star\|_{L^2} + \|m_n^\star - f(x)\|_{L^2}. \qquad (23)$$

The stability bound equation 22 implies

$$\|\widehat{m}_n - m_n^\star\|_{L^2} \leq \|\Gamma\Delta\phi_n\|_{L^2}$$
$$= \left(\mathbb{E}[\Gamma^2\Delta\phi_n^2]\right)^{1/2} \to 0, \qquad (24)$$

while the oracle $L^2$ assumption gives $\|m_n^\star - f(x)\|_{L^2} \to 0$. Hence, by equation 23 and equation 24, $\|\widehat{m}_n - f(x)\|_{L^2} \to 0$. Since $\widehat{m}_n = \widetilde{\mu}_n(x;\hat{\phi}_n)$, this is exactly

$$\widetilde{\mu}_n(x;\hat{\phi}_n) \to f(x) \quad \text{in } L^2.$$

$\square$

