# OpenReview forum: "Taking the GP Out of the Loop"
_ICML.cc/2026/Conference — ICML 2026 regular_

### Official Review · Reviewer_jLWd · 2026-03-01

**Soundness:** 2
**Presentation:** 2
**Significance:** 3
**Originality:** 3
**Overall Recommendation:** 5
**Confidence:** 4

**Summary:**

This paper addresses black-box optimization, i.e., the optimization of functions of unknown form that can only be observed pointwise and do not provide gradient information. Bayesian optimization (BO) is a popular technique for such problems because it has been shown to be sample-efficient, i.e., achieving good solutions with few function evaluations, which is an important property given that black-box functions are often expensive to evaluate. In this work, black-box functions are assumed to be relatively inexpensive to evaluate, allowing a larger evaluation budget of several thousand evaluations.

BO operates by iteratively suggesting new points to evaluate and refining a surrogate model of the unknown black-box function $f$. Since the common Gaussian process (GP) surrogate scales cubically (or, as noted by the authors, almost quadratically in modern implementations) with the number of function evaluations, it can be prohibitively slow on large-scale problems, requiring alternative approaches such as _sparse_ GPs or other surrogate models.

This work proposes an alternative surrogate model for large-scale problems: epistemic nearest neighbors (ENN). Inspired by the TuRBO, a method for large-scale BO that operates in occasionally reset _trust regions_, the proposed method proposes candidates to evaluate within a TR and estimates the mean and variance of those candidates using the mean and variance of the closest candidates.

The method scales linearly in the number of observations. To propose candidates, points are chosen randomly from a Pareto set of points with high mean or variance.

The method is evaluated on several benchmarks in scenarios with "natural" and "frozen" noise. While ENN does not outperform TuRBO in most cases, it is considerably faster due to its linear scaling.

Finally, the authors show that ENN is a so-called pseudo-BO method that enjoys certain convergence properties.

**Compliance With Llm Reviewing Policy:**

Affirmed.

**Final Justification:**

The rebuttal addressed concerns regarding the evaluation, and I believe that this work has the potential to have a high impact in the field of BO. Thus, I decided to raise my score.

**Key Questions For Authors:**

* How much of the performance is attributed to the surrogate model and how much to the acquisition function maximization technique?
* Do you assume that you observe $s_m$? If so, how realistic is that in practice? If not, it is unclear to me how $\mu$ and $\sigma$ are calculated.
* Why don't runs start from the same value? Can't we condition on a small, fixed initial set across optimizers?

**Limitations:**

The paper is lacking an impact statement.

**Strengths And Weaknesses:**

**Strengths**

While not extensively motivated in this work, BO methods that scale to large evaluation numbers are a highly relevant field of research, since the explicit modeling of uncertainty may outperform existing meta-heuristics such as evolutionary methods. This work proposes a surrogate model that scales linearly in the number of function evaluations and empirically shows fast proposal times. The motivation for the method is clear and reasoned.

The paper is generally well-written and follows a logical structure. The paper is honest about its contributions and does not overstate them.

In addition to ENN, the paper proposes a new method (TuRBO-zero), which does not use TuRBO's surrogate model but simply samples from a set of candidates. This baseline is interesting as, to a certain degree, it allows assessing the impact of the surrogate model vs. the impact of locality.

**Weaknesses**

* The role of the noise is unclear. Section 4.1 reads like $s_m$ is observed when evaluating $f$ - which might not be possible in practice. In contrast, Algorithm 1 does not state this anywhere. Here, the description needs clarification.
* In "Combining estimates" (p.4), $\sum_i^K \sigma^{-2}(x\mid x_i,y_i,s_i)$ should cancel out for both $\mu(x)$ and $\sigma_a^2(x)$ . There seems to be some error in the description of the mean and variance.
* $P$ and $K$ are important hyperparameters, but there are no ablations studying their impact.
* The set of baselines is relatively small. The authors state that they abstain from comparing to random forests due to fitting difficulties, but they seem to be a relevant and necessary baseline for this work.
* For the high-dimensional BO task (Lasso-DNA), recent high-dimensional techniques (e.g., [1] or [2]) are competitive even though they use fewer observations. Arguably, they should be included as baselines with a cap of perhaps 2000 iterations to study how competitive SOTA exact methods are.
* Optuna's behavior on Fig 4 (right) seems odd. Is there a bug?
* The curves for different optimizers do not start from the same initial value. I'd expect that all optimizers are initialized with a fixed set of initial points.

**Less important**

* A few references for suitable problems for large-scale BBO techniques would be good in the introduction.
* The point that "the number of observations necessary to locate a good design depends on aspects of a problem beyond just $D$" currently is vague and should be further discussed by what those aspects are.
* In "Trust Region BO" in "Related Work", the authors write that "[TuRBO] restricts Thompson samples to within a trust region, a small subset of the overall design space where good designs are most likely, thus avoiding needless evaluations elsewhere." I'd agree that this is a technique to lower the wall-clock time if the number of candidates was lower in smaller trust regions but that's not the case for TuRBO. The resolution shrinks for larger TR, thus this isn't really a feature that reduces wall-clock time.
* To my knowledge, Lasso-DNA, as implemented in LassoBench, is not noisy in the sense that re-evaluating the same point gives different function values.
* The optimization of the acquisition function is unconventional, arguably motivated by the fact that $\mu$ and $\sigma$ lie on different scales. However, this conflates the impact of the AF optimization with that of the surrogate model.

*Minor comments*

* p.3, right column, line 147: "see 4" should be "see section 4"
* Line 169, p.4, left column: here, function evaluations may be noiseless, but that wasn't the case in sec 2.1. It would be good to make this consistent.
* Rashidi isn't a good reference for RAASP sampling - they only invented the term "RAASP" but the original method comes from "Regis, Rommel G., and Christine A. Shoemaker. "Combining radial basis function surrogates and dynamic coordinate search in high-dimensional expensive black-box optimization." _Engineering Optimization_ 45.5 (2013): 529-555."
* Fig 1: It would be good to show the proposal time for CMA-ES and TuRBO-zero.

[1] Papenmeier, Leonard, Luigi Nardi, and Matthias Poloczek. "Bounce: Reliable high-dimensional bayesian optimization for combinatorial and mixed spaces." Advances in Neural Information Processing Systems 36 (2023): 1764-1793.
[2] Hvarfner, Carl, Erik O. Hellsten, and Luigi Nardi. "Vanilla Bayesian optimization performs great in high dimensions." Proceedings of the 41st International Conference on Machine Learning. 2024.

---

> ### Author Rebuttal · Authors · 2026-03-31
>
> Thank you for the thoughtful and helpful comments.
>
> ### How much of the performance is attributed to the surrogate model and how much to the acquisition function maximization technique?
> - We are adding extra variations in our experiments, TuRBO with GP and NDS (frozen noise) and TuRBO with GP and UCB (natural noise), to make this attribution possible.
>
> ### Do you assume that you observe s_m? If so, how realistic is that in practice? If not, it is unclear to me how mu and se  are calculated.
> - No. The aleatoric noise may be supplied or inferred. s_m may be observed in any experiment (or simulation) where the noisy objective is observed multiple times. The experimenter would report y = mean(objective values), s_m ‎ =  std(objective values) / sqrt(number of observations), i.e. the standard error of the mean.
> - In the case where the objective is noisy but s_m is not supplied, we set s_m‎ = 0. The aleatoric noise is then just s_0 (inferred by likelihood maximization). We will state "s_m ‎ =  0" explicitly in the paper.
>
> ### Why don't runs start from the same value? Can't we condition on a small, fixed initial set across optimizers?
> - Different optimization algorithms prescribe different initialization methods. Our view was that if we were to modify the initialization methods, that would constitute a change to the algorithm to which we were comparing, thus calling into question whether we had inadvertently harmed our baselines, exaggerating the quality of our own work.
> - While it might be interesting and instructive to study various modifications and ablations of cma, optuna, and TuRBO, they have been studied in many papers with their default initializations, hyperparameters, surrogates, samplers, and acquisition functions, and to change any of component would, in our view, motivate a new comparative study.
>
>
> ### To my knowledge, Lasso-DNA, as implemented in LassoBench, is not noisy in the sense that re-evaluating the same point gives different function values.
> - Lasso-DNA shuffles data points randomly, which introduces noise into the objective. The class `RealBenchmark` takes a seed for this purpose. If unspecified, the seed defaults to 42, not to None (as is typical in many numerical libraries).
>
> ### In "Combining estimates" (p.4), [...] should cancel out for both \mu and \sigma_a^2. There seems to be some error in the description of the mean and variance.
> - $mu(x) = \frac{ \sum_i^K \sigma^{-2}(x \mid x_i,y_i,s_i) \mu(x \mid x_i, y_i, s_i) }{ \sum_i^K \sigma^{-2}(x \mid x_i,y_i,s_i) }$ The expression is of the form $\frac{ \sum_i^K w_i x_i } {\sum_i^K w_i}$, i.e. a weighted sum over a normalizer.
>
> ### The set of baselines is relatively small. The authors state that they abstain from comparing to random forests due to fitting difficulties, but they seem to be a relevant and necessary baseline for this work.
> - Will will add SMAC and DNGO to the experiments where computation time is not prohibitive.
> ```
> N=1000, D=100, Sphere (unimodal)
> Surrogate           Fit t (s)       t/t_ENN          NRMSE         LogLik
> ---------------------------------------------------------------------------
> ENN            6.25e-04 ± 1.7e-05         1   1.02681  ± 0.0006  797.65 ± 1
> SMAC RF        1.36649  ± 0.057      2187.6   1.00078  ± 0.0012  825.78 ± 2
> DNGO          75.9242   ± 1.3      121546.1   1.01224  ± 0.0052  800.01 ± 12
> Exact GP     166.773    ± 8.5      266984.3   1.03072  ± 0.0039  774.18 ± 6
> SVGP_default   9.81392  ± 0.24      15711.0   0.99302  ± 0.00055 802.37 ± 2
> SVGP_linear    2.68122  ± 0.084      4292.3   0.993904 ± 0.0004  820.94 ± 2
> Vecchia        4.07247  ± 0.1        6519.6   1.00168  ± 0.00048 824.64 ± 1
> ```
>
> ### Optuna's behavior on Fig 4 (right) seems odd. Is there a bug?
> - Yes, a plotting bug. Optuna was omitted from frozen-noise Hopper because the per-replication running time exceeded our 5-hour limit. See note at the bottom-left of pg. 6 (line 327) and the "-" marker in Table 1.
>
> ###  P and K are important hyperparameters, but there are no ablations studying their impact.
> - Yes. We removed K ablations for space but will add them to the appendix along with ablations for P. The method proves robust to changes in K & P.
>
> ### The optimization of the acquisition function is unconventional, arguably motivated by the fact that mu and sigma lie on different scales. However, this conflates the impact of the AF optimization with that of the surrogate model.
> - This concern may be somewhat mitigated by our presentation of the frozen-noise version which does not use the fitting procedure.

---

> > ### Author Rebuttal · Reviewer_jLWd · 2026-04-01
> >
> > The rebuttal resolved most concerns. Adding experiments like "TuRBO with GP and NDS (frozen noise) and TuRBO with GP and UCB (natural noise)" is a good idea and will help in understanding the contribution of the different components. Also, the additional ablation studies, details, and baselines will improve the paper. However, those additional experiments probably exceed the scope of a rebuttal. Thus, I leave my score as is.

---

> > > ### Author Response · Authors · 2026-04-04
> > >
> > > The table below shows the cumulative time spent generating proposals (fitting surrogate + acquisition, excluding running the simulator for function evaluation) for all of the optimization methods discussed. We will update the plots and Table 1 so that where a method can complete a run in less than 5 hours the data will be presented otherwise we will note the failure to complete within the time allotted. Making this explicit and displaying it side-by-side in a single table should communicate our findings more clearer to the reader.
> > >
> > > | opt_name | tlunar:fn | tlunar | push:fn | push | hop:fn | hop | bw-heur:fn | bw-heur |
> > > |---|---|---|---|---|---|---|---|---|
> > > | ucb | 2217.5 | **> 5hr** | **> 5hr** | **> 5hr** | **> 5hr** | **> 5hr** | 10541.8 | **> 5hr** |
> > > | lei | **> 5hr** | **> 5hr** | **> 5hr** | **> 5hr** | **> 5hr** | **> 5hr** | **> 5hr** | **> 5hr** |
> > > | smac | 461.6 | 13031.8 | 15296.1 | 15364.6 | 14648.8 | 17056.4 | 1382.3 | 15630.6 |
> > > | dngo | 364.6 | **> 5hr** | 16893.3 | **> 5hr** | **> 5hr** | **> 5hr** | 2566.3 | **> 5hr** |
> > > | vecchia | 111.5 | **> 5hr** | 4336.6 | **> 5hr** | 4685.2 | **> 5hr** | 491.5 | **> 5hr** |
> > > | ucb:Msparse | **> 5hr** | **> 5hr** | **> 5hr** | **> 5hr** | **> 5hr** | **> 5hr** | **> 5hr** | **> 5hr** |
> > > | turbo-enn | 0.5 | 343.3 | 19.1 | 461.6 | 39.4 | 1402.7 | 1.7 | 543.9 |
> > > | turbo-one | 47.7 | 3119.6 | 946.8 | 3397.6 | 1734.7 | 10595.0 | 145.1 | 3265.9 |
> > >
> > >
> > > For K & P ablation data, please see:
> > >
> > > - [ackley_D=10_N=1000_sweep_k.png](https://github.com/dsweet99/anon/blob/main/ackley_D%3D10_N%3D1000_sweep_k.png): Fits of ENN to N=1000 samples of 10-D Ackley at various K. We see optimal mean prediction quality (in NRMSE and LogLik) at K=10.
> > > - [ackley_D=10_N=1000_sweep_p_fit.png](https://github.com/dsweet99/anon/blob/main/ackley_D%3D10_N%3D1000_sweep_p_fit.png): Fits of ENN to N=1000 samples of 10-D Ackley at various P. Prediction quality (NRMSE, LogLik) improves with increasing P and stabilizes at P=10. Uncertainty in LogLik decreases with increasing P, as discussed in section 4.1.
> > > - [sphere_D=10_N=1000_sweep_k.png](https://github.com/dsweet99/anon/blob/main/sphere_D%3D10_N%3D1000_sweep_k.png): Fits of ENN to N=1000 samples of 10-D Sphere at various K. We see optimal mean prediction quality (in NRMSE and LogLik) at K=10.
> > > - [sphere_D=10_N=1000_sweep_p_fit.png](https://github.com/dsweet99/anon/blob/main/sphere_D%3D10_N%3D1000_sweep_p_fit.png): Fits of ENN to N=1000 samples of 10-D Ackley at various P. LogLik (mostly) improves with increasing P Uncertainty in LogLik decreases with increasing P, as discussed in section 4.1.
> > > - [turbo_enn_tlunar_fn_sweep_k.png](https://github.com/dsweet99/anon/blob/main/turbo_enn_tlunar_fn_sweep_k.png): K is a very stable hyperparameter on this problem. (Other runs on pure functions show variation with a maximum in K.)
> > > - [turbo_enn_tlunar_sweep_p.png](https://github.com/dsweet99/anon/blob/main/turbo_enn_tlunar_sweep_p.png): P is a very stable hyperparameter on this problem.

---

### Official Review · Reviewer_uiVs · 2026-03-03

**Soundness:** 3
**Presentation:** 3
**Significance:** 3
**Originality:** 2
**Overall Recommendation:** 4
**Confidence:** 3

**Summary:**

This paper addresses the challenge of Bayesian Optimization with Many Observations (BOMO) by proposing TuRBO-ENN, a method that integrates a lightweight K-Nearest Neighbors (KNN) surrogate model into the TuRBO framework. The core contribution is the replacement of the standard Gaussian Process (GP) with an Epistemic Neural Neighbor (ENN) model, which estimates function values and decomposes uncertainty (aleatoric and epistemic) directly from $K$ nearest observations. This approach aims to reduce the proposal time complexity to $\mathcal{O}(N)$. Regarding acquisition strategies, the authors employ a fitting-free non-dominated sorting (NDS) approach for deterministic settings and a UCB-based approach for noisy settings. Theoretically, the paper validates the surrogate model, uncertainty quantification, and acquisition mechanism under the "Pseudo-Bayesian Optimization (PBO)" framework. Empirically, TuRBO-ENN is shown to reduce proposal time by one to two orders of magnitude compared to the original TuRBO—exhibiting linear $\mathcal{O}(N)$ scaling—while maintaining competitive optimization performance.

**Compliance With Llm Reviewing Policy:**

Affirmed.

**Final Justification:**

The author's rebuttal effectively addressed my original concern, so I have raised my score to 4 (from the original 3).

**Key Questions For Authors:**

1. Could you provide ablation studies showing how sensitive the model's performance is to variations in $K$ and $P$? Specifically, why was $K$ fixed to 10, and how might different values affect the overall results?
2. How robust is the algorithm's performance and uncertainty quantification to estimation errors in $c_e$ and $s_0$? Please clarify the potential consequences of deviations caused by using an approximate likelihood function.
3. Since the core contribution is a lightweight non-GP surrogate, why are mainstream baselines like Random Forest (SMAC) or Neural Networks (DNGO) omitted? Furthermore, could you elaborate on how your KNN surrogate fundamentally differs from previously existing KNN-based methods?
4. How can the claims regarding large-scale data performance be justified when Figure 1 only scales up to $N=1000$? Additionally, please explain why Vecchia GP is depicted as slower than exact GP, which contradicts the standard computational advantages of Vecchia approximations.

**Limitations:**

Yes.

**Strengths And Weaknesses:**

## Strengths
* The ENN is a simple yet well-motivated surrogate that provides a mean predictor and decomposed uncertainty (aleatoric and epistemic) with fitting and querying costs that are linear in $N$.
* The Pseudo-Bayesian Optimization (PBO) analysis provides a valuable theoretical lens for convergence in the noiseless setting. This is a notable strength, as such theoretical rigor is often absent in BO algorithms that rely on non-GP surrogate models.
* The discussion regarding noise handling and the evaluation protocol (specifically, passive denoising for natural noise) is comprehensive and carefully explained.
## Weaknesses
* There are no ablation studies regarding the hyperparameters $K$ and $P$. The number of neighbors $K$ is fixed to 10 throughout the paper without justification. It is unclear how sensitive the model's performance is to variations in $K$.
* The impact of $c_e$ and $s_0$ on performance and uncertainty quantification is not sufficiently demonstrated. The authors use an approximate likelihood function to estimate these parameters, which implies a risk of deviation from true values. The potential consequences of such deviations on the algorithm's robustness are not adequately addressed.
* The evaluation lacks comparisons with mainstream non-GP surrogate baselines. Given that the paper's core contribution is a lightweight non-GP surrogate (ENN) designed to bypass GP computational bottlenecks, it is critical to compare against established non-GP methods such as Random Forest-based SMAC[1] or Neural Network-based DNGO[2]. Furthermore, considering that the KNN surrogate is not first proposed in this paper, a more detailed discussion of its distinctions from previous methods is needed.
* The experiments in Figure 1 are insufficient to substantiate claims regarding performance on large-scale data. The x-axis (number of observations $N$) only extends to $N=1000$. Furthermore, Figure 1 depicts Vecchia GP methods as slower than traditional exact GPs, which raises questions about the implementation or the scale of the data used, as Vecchia methods typically excel in larger regimes.

[1] Hutter, Frank, Holger H. Hoos, and Kevin Leyton-Brown. "Sequential model-based optimization for general algorithm configuration." In International conference on learning and intelligent optimization, pp. 507-523. Berlin, Heidelberg: Springer Berlin Heidelberg, 2011.

[2] Snoek, Jasper, Oren Rippel, Kevin Swersky, Ryan Kiros, Nadathur Satish, Narayanan Sundaram, Mostofa Patwary, Mr Prabhat, and Ryan Adams. "Scalable bayesian optimization using deep neural networks." In International conference on machine learning, pp. 2171-2180. PMLR, 2015.

---

> ### Author Rebuttal · Authors · 2026-03-31
>
> ### Could you provide ablation studies showing how sensitive the model's performance is to variations in K and P? Specifically, why was  fixed to 10, and how might different values affect the overall results?
> - Yes. We removed K ablations for space, but could add them to the appendix along with ablations for P. The method proves robust to changes in K & P.
>
> ### How robust is the algorithm's performance and uncertainty quantification to estimation errors in c_0 and s_0? Please clarify the potential consequences of deviations caused by using an approximate likelihood function.
> - Since estimation error in c_0 and s_0 decreases with P, this would be answered, in part, by the ablation of fitting quality vs. P.
>
> ### Since the core contribution is a lightweight non-GP surrogate, why are mainstream baselines like Random Forest (SMAC) or Neural Networks (DNGO) omitted? Furthermore, could you elaborate on how your KNN surrogate fundamentally differs from previously existing KNN-based methods?
> - SMAC and DNGO have many hyperparameters that, themselves, need tuning. and can take a long time to fit leading to prohibitive computation times. That being said, we will add SMAC and DNGO to the subset of problems where running them is feasible and will report in Table 1 which method, environment pairs cannot complete in under 5 hours.
> - You can see sample fitting times for SMAC and DNGO surrogates and how they compare to other surrogates:
> ```
> N=1000, D=10, Ackley (multimodal)
> Surrogate                 Fit t (s)      t/t_ENN          NRMSE              LogLik
> ---------------------------------------------------------------------------------------
> ENN                  0.000488 ± 1.1e-05        1   0.858375 ± 0.011      -1357.89 ± 360
> SMAC RF              0.132719 ± 0.0012     272.2    0.91347 ± 0.003      -2342.64 ± 180
> DNGO                48.0451   ± 0.47     98521.2   0.858092 ± 0.017      -1018.43 ± 200
> Exact GP             4.48871  ± 0.22      9204.6   0.700453 ± 0.0058      -262.89 ±  20
> SVGP_default         8.11583  ± 0.11     16642.3   0.879975 ± 0.0037     -1290.43 ±  86
> SVGP_linear          1.92635  ± 0.048     3950.2   0.914502 ± 0.0024      -964.92 ±  59
> Vecchia              2.46467  ± 0.063     5054.1   0.758656 ± 0.0065      -468.52 ±  28
> ```
>
> ```
> N=1000, D=100, Sphere (unimodal)
> Surrogate           Fit t (s)       t/t_ENN          NRMSE         LogLik
> ---------------------------------------------------------------------------
> ENN            6.25e-04 ± 1.7e-05         1   1.02681  ± 0.0006  797.65 ± 1
> SMAC RF        1.36649  ± 0.057      2187.6   1.00078  ± 0.0012  825.78 ± 2
> DNGO          75.9242   ± 1.3      121546.1   1.01224  ± 0.0052  800.01 ± 12
> Exact GP     166.773    ± 8.5      266984.3   1.03072  ± 0.0039  774.18 ± 6
> SVGP   9.81392  ± 0.24      15711.0   0.99302  ± 0.00055 802.37 ± 2
> Vecchia        4.07247  ± 0.1        6519.6   1.00168  ± 0.00048 824.64 ± 1
> ```
>
> ### How can the claims regarding large-scale data performance be justified when Figure 1 only scales up to 1000? Additionally, please explain why Vecchia GP is depicted as slower than exact GP, which contradicts the standard computational advantages of Vecchia approximations.
> - Even at N=1000, GP methods are >60x slower *and* their times are scaling like O(N^2) (rather than ENN's O(N)), thus the gap in running time would only widen as N increased to 10's of thousands.
> - Note that although TuRBO uses a GP, its trust-region controller occasionally "resets", meaning that it discards all observations and restarts the collection process. This means that even as the number of function evaluations increases, the number of samples used to fit the GP may not. This is how TuRBO scales to large N.
> - Vecchia can be slower than exact GP.
> - i) The hyperparameter fitting for either an exact GP or Vecchia is an optimization that iterates until some convergence criterion is met (Vecchia: relative improvement, GP (BoTorch): tolerance), so the number of iterations can be higher or lower for Vecchia.
> - (ii) Vecchia picks neighbors and applies an ordering constraint, which  an exact GP fit does not.
> - (iii) Slight scaling complications. Vecchia scales as O(N * m^3) where m is the number of neighbors, whereas exact GP scales as O(N^3).  Our paper points out that GPyTorch uses BMMM to reduce the cube to a square, making these O(N*m^2) and O(N^2), respectively. In the case where m > sqrt(N), then Vecchia is slower than an exact GP.  Using the VecchiaGP repo's recommendation of `m = int(7.2 * np.log10(n) ** 2)` puts the minimum N for which Vecchia is (in scaling) faster than exact GP at (int(7.2 * np.log10(n) ** 2))**2 = N_cutoff ~= 16,333.
> - In practice N_cutoff might vary greatly due to (i) and (ii) as well as the exact details of the Vecchia and GP fitting routines and the nature of the data being fit (which might lead to more of fewer iterations).

---

> > ### Author Rebuttal · Reviewer_uiVs · 2026-04-03
> >
> > I thank the authors for their response and the additional experimental results. I acknowledge the following points:
> >
> > 1. The supplementary surrogate comparison table (SMAC RF, DNGO, etc.) provides useful context on fitting times and surrogate quality. However, I note that the metrics NRMSE and LogLik are not explicitly defined in the rebuttal, which makes it difficult to fully assess the results.
> >
> > 2. The technical explanation for why Vecchia GP can be slower than exact GP at N=1000 is reasonable. I accept that at small N, the overhead of neighbor selection and the relationship between m and sqrt(N) can make Vecchia less efficient.
> >
> > However, several key concerns remain unresolved:
> >
> > (a) Regarding Vecchia GP as a baseline: The authors' own analysis shows that Vecchia's computational advantage materializes at approximately N > 16,000. Since the paper positions itself for BO scenarios with many observations, Vecchia is arguably the most relevant nearest-neighbor-based baseline in exactly the regime the paper targets. Excluding Vecchia from the main experiments based solely on N=1000 results — a scale where it is not expected to excel — undermines the convincingness of the main experimental evaluation. I would expect to see comparisons at larger N where Vecchia is competitive.
> >
> > (b) Regarding ablations on P: The authors indicated that ablations on P would be added, and that these would also address the sensitivity of c_0 and s_0 estimation. However, no such results were provided in the rebuttal. This concern remains open.
> >
> > (c) Regarding ablations on K: Similarly, no concrete results were provided beyond a statement that the method is robust.
> >
> > Overall, while I appreciate the authors' efforts, the rebuttal does not sufficiently address my main concerns regarding experimental rigor and the lack of ablation studies. I maintain my current score.

---

> > > ### Author Response · Authors · 2026-04-04
> > >
> > > > NRMSE and LogLik are not explicitly defined in the rebuttal,
> > >
> > > We fit on $N=1000$ points with $x \sim Uniform[ [0,1]^{D} ]$ then compute $NRMSE$ and $LogLik$ on a second (i.e., out-of-sample) set of $N=1000$ similarly-generated points.
> > >
> > > $NRMSE(y, \hat{y}) = \frac{ \sum_{i=1}  (y_i - \hat{y}_i )^2 }{ \sum_{i=1} y_i^2 }  $
> > >
> > > and $LogLik(y, \hat{y}, v) = \sum_{i=1}^n \left[ -\frac12 \log(2\pi v_i) - \frac{(y_i - \hat{y}_i)^2}{2 v_i} \right]$ where $\hat{y_i}$  and $v_i$ are the mean and variance values generated by the surrogate. $NRMSE$ is, therefore, sensitive only to the mean predictions, whereas $LogLik$ is sensitive to both the mean and variance predictions.
> > >
> > > > The authors' own analysis shows that Vecchia's computational advantage materializes at approximately N > 16,000. Since the paper positions itself for BO scenarios with many observations, Vecchia is arguably the most relevant nearest-neighbor-based baseline in exactly the regime the paper targets.
> > >
> > > The difference we need to highlight is between a one-shot regression task with a fixed $N$ and a sequential optimization. To get to N=16,000, an optimization must pass through N=1, 2, 3, ... where Vecchia can be even slower than an exact GP.  At every round a new surrogate is fit. We are concerned with $\sum_i t_{\text{proposal,i}} $, the cumulative proposal time.
> > >
> > > The table below shows that the time spent *just on generating proposals* (fitting surrogate + acquisition, excluding running the simulator for function evaluation) in many of these tasks exceeds 5 hours (18,000 seconds) for Vecchia, whereas the times for TuRBO ENN are significantly lower.   We originally omitted Vecchia, SMAC, DNGO, UCB, sparse GP with UCB, and LEI (Log Expected Improvement) from our numerical experiments because of their exceedingly long running times compared even to standard TuRBO. We included Optuna even though its running time exceeded 5 hours on one of the Hopper problems because it did not "time out" on any other task.. That being said, our revision will present a larger table and make this all explicit. (Note that while SMAC does not quite exceed 18,000 seconds on any task, it is very close on most tasks.)
> > >
> > > | opt_name | tlunar:fn | tlunar | push:fn | push | hop:fn | hop | bw-heur:fn | bw-heur |
> > > |---|---|---|---|---|---|---|---|---|
> > > | ucb | 2217.5 | **> 5hr** | **> 5hr** | **> 5hr** | **> 5hr** | **> 5hr** | 10541.8 | **> 5hr** |
> > > | lei | **> 5hr** | **> 5hr** | **> 5hr** | **> 5hr** | **> 5hr** | **> 5hr** | **> 5hr** | **> 5hr** |
> > > | smac | 461.6 | 13031.8 | 15296.1 | 15364.6 | 14648.8 | 17056.4 | 1382.3 | 15630.6 |
> > > | dngo | 364.6 | **> 5hr** | 16893.3 | **> 5hr** | **> 5hr** | **> 5hr** | 2566.3 | **> 5hr** |
> > > | vecchia | 111.5 | **> 5hr** | 4336.6 | **> 5hr** | 4685.2 | **> 5hr** | 491.5 | **> 5hr** |
> > > | ucb:Msparse | **> 5hr** | **> 5hr** | **> 5hr** | **> 5hr** | **> 5hr** | **> 5hr** | **> 5hr** | **> 5hr** |
> > > | turbo-enn | 0.5 | 343.3 | 19.1 | 461.6 | 39.4 | 1402.7 | 1.7 | 543.9 |
> > > | turbo-one | 47.7 | 3119.6 | 946.8 | 3397.6 | 1734.7 | 10595.0 | 145.1 | 3265.9 |
> > >
> > >
> > > > Regarding ablations on P
> > > > Regarding ablations on K:
> > >
> > > Please see:
> > >
> > > - [ackley_D=10_N=1000_sweep_k.png](https://github.com/dsweet99/anon/blob/main/ackley_D%3D10_N%3D1000_sweep_k.png): Fits of ENN to N=1000 samples of 10-D Ackley at various K. We see optimal mean prediction quality (in NRMSE and LogLik) at K=10.
> > > - [ackley_D=10_N=1000_sweep_p_fit.png](https://github.com/dsweet99/anon/blob/main/ackley_D%3D10_N%3D1000_sweep_p_fit.png): Fits of ENN to N=1000 samples of 10-D Ackley at various P. Prediction quality (NRMSE, LogLik) improves with increasing P and stabilizes at P=10. Uncertainty in LogLik decreases with increasing P, as discussed in section 4.1.
> > > - [sphere_D=10_N=1000_sweep_k.png](https://github.com/dsweet99/anon/blob/main/sphere_D%3D10_N%3D1000_sweep_k.png): Fits of ENN to N=1000 samples of 10-D Sphere at various K. We see optimal mean prediction quality (in NRMSE and LogLik) at K=10.
> > > - [sphere_D=10_N=1000_sweep_p_fit.png](https://github.com/dsweet99/anon/blob/main/sphere_D%3D10_N%3D1000_sweep_p_fit.png): Fits of ENN to N=1000 samples of 10-D Ackley at various P. LogLik (mostly) improves with increasing P Uncertainty in LogLik decreases with increasing P, as discussed in section 4.1.
> > > - [turbo_enn_tlunar_fn_sweep_k.png](https://github.com/dsweet99/anon/blob/main/turbo_enn_tlunar_fn_sweep_k.png): K is a very stable hyperparameter on this problem. (Other runs on pure functions show variation with a maximum in K.)
> > > - [turbo_enn_tlunar_sweep_p.png](https://github.com/dsweet99/anon/blob/main/turbo_enn_tlunar_sweep_p.png): P is a very stable hyperparameter on this problem.

---

### Official Review · Reviewer_BVZF · 2026-03-12

**Soundness:** 3
**Presentation:** 3
**Significance:** 2
**Originality:** 2
**Overall Recommendation:** 3
**Confidence:** 3

**Summary:**

This paper is about making Bayesian optimization work when you already have a lot of data. Standard Bayesian optimization usually uses a Gaussian process (GP) as its surrogate model, but GP fitting becomes expensive as the number of observations grows, and that can make the optimizer itself slow. The paper’s core idea is: instead of fitting a GP every round, estimate the value of a new point by looking only at its nearest previously evaluated points, and use their distances to estimate uncertainty.

The proposed surrogate is called Epistemic Nearest Neighbors (ENN). For a candidate point x, ENN takes its K nearest neighbors, combines their observed objective values with inverse-variance weights, and defines uncertainty using two parts: observation noise and “how far am I from past data?” uncertainty. The authors then plug ENN into TuRBO, replacing TuRBO’s GP and Thompson sampling with ENN plus either UCB for noisy problems or a non-dominated sort.

The main contributions are:

- A simple nearest-neighbor surrogate for BO that is much cheaper than a GP at large N.
- A modified optimizer, TuRBO-ENN, with claimed linear-in-N proposal cost under fixed K and fixed likelihood-subsample size P.
- Empirical results showing similar optimization quality to TuRBO but much lower proposal time, including experiments up to 50,000 observations (pp. 6-8).
- An appendix argument that a noiseless, Pareto-compatible version fits the Pseudo-Bayesian Optimization framework and therefore inherits a convergence guarantee.

**Compliance With Llm Reviewing Policy:**

Affirmed.

**Key Questions For Authors:**

- Most of the figures have un-readable fontsizes for labels and ticklables -- can you fix them?
- In Appendix A, the convergence argument is for a noiseless setting and a scalarized Pareto-compatible acquisition, while the main experimental method uses UCB for noisy problems and exact NDS sampling for deterministic ones. Can you clarify precisely which deployed variant is theoretically covered?
- How sensitive are results to the fixed choices of K=10 and the likelihood subsample size P? Did you test adaptive or problem-dependent
K, and if so, how much does performance move?
- Since ENN’s main selling point is that it provides both a mean and uncertainty estimate, can you provide a direct uncertainty-quality analysis: calibration, interval coverage, predictive NLL, or surrogate RMSE compared with GP/Vecchia baselines?
- Why are Vecchia GP and sparse GP included in the timing study but not in the main optimization-quality benchmarks? Those would seem to be the most relevant uncertainty-aware scalable competitors.
- What are the main failure modes of ENN? In particular, how does it behave on problems with strong anisotropy, irrelevant dimensions, constraints, mixed variables, or categorical inputs, where Euclidean nearest-neighbor structure may be much less informative?

**Strengths And Weaknesses:**

## Strengths
- Well-motivated problem and clean idea. The paper targets an important regime: BO with many observations, where GP fitting becomes the bottleneck. The proposed replacement is simple, understandable, and computationally much lighter than GP-based BO.
- The experiments push to large N, including runs up to 50k observations, and the reported proposal-time reductions are substantial: roughly 10x to 150x on the main benchmarks, and one to two orders of magnitude overall (pp. 6-8).
- Practical integration with TuRBO, not just a standalone surrogate. The paper does more than propose a new predictor; it adapts it into a competitive trust-region BO pipeline, including a fitting-free deterministic variant. That makes the contribution more actionable than a purely surrogate-level paper.

## Weaknesses
- The theoretical results does not cleanly match the main algorithm used in experiments. Appendix A assumes noiseless observations and proves the PseudoBO conditions for a Pareto-compatible scalarized acquisition, while the main noisy method uses UCB, and the implemented deterministic method samples from the first Pareto front rather than the scalarized rule (pp. 5, 12-14). So the theory-to-method connection is weaker than the paper suggests.
- The main quality benchmarks omit some of the most relevant scalable BO baselines. Sparse GP and Vecchia-style GP methods appear in the timing plot (Figure 1), but not in the core optimization-quality experiments (Figures 3-7). That makes it harder to judge whether ENN is competitive with the strongest uncertainty-aware scalable alternatives, rather than just with standard TuRBO.
- Several changes are bundled together, so attribution is unclear. Relative to TuRBO, the paper changes the surrogate, the acquisition rule, and in deterministic settings also removes fitting entirely. The surrogate-free ablation helps somewhat, but it still does not isolate how much of the final behavior comes from ENN itself versus UCB/NDS or other simplifications.
- The paper claims uncertainty estimation is a key feature, but does not evaluate it directly. There is no predictive RMSE, calibration, interval coverage, or likelihood-style analysis for $\mu(x)$ and $\sigma(x)$. All evidence is indirect, via downstream optimization curves.
- Benchmark coverage is still narrower than the paper’s motivation. The main tasks are simulated control problems plus one HPO problem, all continuous and box-constrained, with dimensions from 12 to 180. There are no constrained, mixed-variable, categorical, or real physical-system experiments, and no direct stress test of ENN’s Euclidean-neighbor assumption in strongly anisotropic or harder high-dimensional settings.
- I think the paper would be strengthened by a discussion of how the method compares against other scalable GP methods such as SVGP and SAASBO.

---

> ### Author Rebuttal · Authors · 2026-03-31
>
> ### Most of the figures have un-readable fontsizes for labels and ticklables -- can you fix them?
> - Yes.
>
> ### In Appendix A, the convergence argument is for a noiseless setting and a scalarized Pareto-compatible acquisition, while the main experimental method uses UCB for noisy problems and exact NDS sampling for deterministic ones. Can you clarify precisely which deployed variant is theoretically covered?
> We thank the reviewer for raising this point. In the revised Appendix A, we now make the scope of the theory explicit. The formal result is for an **idealized noiseless scalarized ENN procedure** built from the same pseudo-posterior pair \\((\\mu_n,\\sigma_n)\\) used by TuRBO-ENN; it does **not** prove a Chen--Lam convergence theorem for the deployed finite-candidate trust-region procedure as implemented.
>
> Concretely, Appendix A shows that in the noiseless setting the ENN mean \\(\\mu_n\\) satisfies local consistency (LC), the ENN uncertainty \\(\\sigma_n\\) satisfies sequential no-empty-ball (SNEB), and the auxiliary scalarized acquisition
> \\[
> \\widetilde{\\alpha}_n^\\lambda(x)
> = \\lambda(\\mu_n(x)-m_n) + (1-\\lambda)\\sigma_n(x), \\qquad \\lambda\\in(0,1),
> \\]
> satisfies the improvement property (IP). Thus, the appendix verifies the SP/UQ/AF conditions in the Chen--Lam PseudoBO framework for this auxiliary noiseless scalarized ENN construction.
>
> For the **deterministic deployed variant**, Appendix A proves a narrower geometric result: exact NDS on \\((\\mu_n,\\sigma_n)\\) is related to the same scalarized/UCB family on the supported portion of the first Pareto front. However, the implemented uniform first-front sampler is **not** itself given a Chen--Lam convergence theorem.
>
> For the **noisy deployed variant**, Appendix A is explicit that it is not covered by the present theorem. The noisy experiments use
> \\[
> UCB(x)=\\mu(x)+\\sigma(x),
> \\]
> which is already a fixed scalarization of the same pseudo-posterior pair.
>
>
> ### How sensitive are results to the fixed choices of K=10 and the likelihood subsample size P? Did you test adaptive or problem-dependent K, and if so, how much does performance move?
> - We removed K ablations for space, but will add them to the appendix along with an ablation of P. The method proves robust to changes in K & P. We tested experimented with K in the fit, but it added to the running time w/o impacting performance.
>
> ### Since ENN’s main selling point is that it provides both a mean and uncertainty estimate, can you provide a direct uncertainty-quality analysis: calibration, interval coverage, predictive NLL, or surrogate RMSE compared with GP/Vecchia baselines?
> - Yes. We will add to the appendix comparisons of fitting quality and speed between ENN and GP, Sparse GP, Vecchia GP, tree models (SMAC), and neural networks (DNGO), like this:
> ```
> N=1000, D=10, Ackley (multimodal)
> Surrogate                 Fit t (s)      t/t_ENN          NRMSE              LogLik
> ---------------------------------------------------------------------------------------
> ENN                  0.000488 ± 1.1e-05        1   0.858375 ± 0.011      -1357.89 ± 360
> SMAC RF              0.132719 ± 0.0012     272.2    0.91347 ± 0.003      -2342.64 ± 180
> DNGO                48.0451   ± 0.47     98521.2   0.858092 ± 0.017      -1018.43 ± 200
> Exact GP             4.48871  ± 0.22      9204.6   0.700453 ± 0.0058      -262.89 ±  20
> SVGP        8.11583  ± 0.11     16642.3   0.879975 ± 0.0037     -1290.43 ±  86
> Vecchia              2.46467  ± 0.063     5054.1   0.758656 ± 0.0065      -468.52 ±  28
> ```
>
> ### Why are Vecchia GP and sparse GP included in the timing study but not in the main optimization-quality benchmarks? Those would seem to be the most relevant uncertainty-aware scalable competitors.
> - The timing study shows that they are very slow in wall time. They cannot meet our 5-hour, single-run cutoff for most simulators, however we will add them to the figures where they do and will report in Table 1 which method, environment pairs cannot complete in 5 hours
>
> ### What are the main failure modes of ENN? In particular, how does it behave on problems with strong anisotropy, irrelevant dimensions, constraints, mixed variables, or categorical inputs, where Euclidean nearest-neighbor structure may be much less informative?
> - With regards to this paper, we consider these to be interesting topics for future study.
> - Strong anisotropy and irrelevant dimensions could be problematic since we do not tune any per-dimension hyperparameters. At the very least this represents an opportunity for improvement. We discuss this in section 6, lines 424-430.
> - But to answer the reviewer directly: We have run preliminary, future-oriented ENN studies with per-dimension weights, mixed variables, parameter constraints, multiple constrained & unconstrained metrics, and contextual components with promising results. We look forward to the opportunity to flesh this work out. We discuss these topics in section 6, lines 436-439.

---

> > ### Author Rebuttal · Reviewer_BVZF · 2026-04-03
> >
> > I thank the authors for responding to my review comments. The question on theoretical statement has been addressed. For other questions, the authors have promised to address them in the final paper.

---

### Official Review · Reviewer_fWyZ · 2026-03-13

**Soundness:** 2
**Presentation:** 4
**Significance:** 2
**Originality:** 3
**Overall Recommendation:** 4
**Confidence:** 4

**Summary:**

This paper explores the scaling of Bayesian optimisation to settings with many observations  by replacing Gaussian process surrogates with a lightweight K-nearest-neighbour approach called Epistemic Nearest Neighbours (ENN). The paper proposes TuRBO-ENN, which integrates ENN into the TuRBO framework, achieving O(N) proposal time compared to O(N^2) for GP-based methods.

**Compliance With Llm Reviewing Policy:**

Affirmed.

**Final Justification:**

Thanks for the engaging rebutall. I agree with your comments regarding GPU/CPU e.t.c

I raise my score.

**Key Questions For Authors:**

Can you provide direct comparisons with EI-based acquisition on at least a subset of problems?

Why were fast GP baselines (particularly Vecchia or inducing point methods) excluded from the main benchmark tables?

Can you demonstrate calibration of your uncertainty estimates through dedicated modelling experiments?

How sensitive is performance to the choice of K? What happens with K ∈ {5, 20, 50}?

Can you include an ablation with GP+NDS to separate model effects from acquisition effects?

**Limitations:**

Yes

**Strengths And Weaknesses:**

1. My major concern is a lack of comparisons to fast GP methods. Significant work has been invested in speeding up GPs, particularly for BO (e.g. inducing point methods, Vecchia, and the refs at the bottom of this review). You discuss some methods in Section 3, but these are never empirically compared against, which is surprising given that speed is one of your main arguments for the method.

2. I am also concerned that you never try other alternatives to GPs. The main argument of you paper is that you speed up BO by changing the surrogate model. In order to justify this, you need to compare with other "faster" surrgates, like work using Random forests and ensembles of NNs e.t.c.

3. It is difficult to justify that this approach "has principled UQ", as the construction is quite ad hoc. Evidence of calibration would be highly convincing—modelling-only experiments (without optimisation) would demonstrate whether the uncertainty estimates are reliable. GPs benefit from well-understood priors, which contribute to BO's sample efficiency. In contrast, ENN relies on a Euclidean distance metric. Also, please can you clarify what Line 215: "treating the observations as independent" means.

4. UCB is known to perform poorly compared to more sophisticated acquisition functions (primarily due to the need to tune beta). Whilst I appreciate that UCB provides computational speedups in your setup, it would be valuable to quantify how much performance you are sacrificing compared to Expected Improvement (EI), for example. I guess one issue would be that you posterior is non-smooth, which could prove problematic for acquisition optimization. This should be discussed more in the paper.


Maus, Natalie, et al. "Approximation-aware bayesian optimization." Advances in Neural Information Processing Systems 37 (2024): 21114-21140.

Vakili, Sattar, et al. "Scalable Thompson sampling using sparse Gaussian process models." Advances in neural information processing systems 34 (2021): 5631-5643.

Lin, Jihao Andreas, et al. "Sampling from Gaussian process posteriors using stochastic gradient descent." Advances in neural information processing systems 36 (2023): 36886-36912.

---

> ### Author Rebuttal · Authors · 2026-03-31
>
> ### Can you provide direct comparisons with EI-based acquisition on at least a subset of problems?
> - Yes. We will add them.
>
> ### Why were fast GP baselines (particularly Vecchia or inducing point methods) excluded from the main benchmark tables?
> - They are only fast in the scaling sense, not in wall-time. They were too slow to be usable on the large-scale problems we studied. See Figure 1 and also the table below compares fitting times for various surrogates, including Vecchia.  We will add Vecchia for a subset of the problems where a single optimization can run in under 5 hours and will report in Table 1 which method, environment pairs cannot complete in 5 hours.  Also we will include the table below in the appendix.
>
> ### Can you demonstrate calibration of your uncertainty estimates through dedicated modelling experiments?
> - Yes. See sample table below. We will include that and more examples for other N, D, and function types (e.g., unimodal, multimodal, long valley, etc.)
>
> ### How sensitive is performance to the choice of K? What happens with K ∈ {5, 20, 50}?
> - This ablation was cut for space. We will include it in the appendix along with an ablation of P. The method proves robust to changes in K & P.
>
> ### Can you include an ablation with GP+NDS to separate model effects from acquisition effects?
> - This is a good idea. We will compare TuRBO+GP+NDS to TuRBO+ENN+NDS. We will also run TuRBO+GP+UCB to compare to TuRBO+ENN+UCB.
>
>
> ```
> N=1000, D=10, Ackley (multimodal)
> Surrogate                 Fit t (s)      t/t_ENN          NRMSE              LogLik
> ---------------------------------------------------------------------------------------
> ENN                  0.000488 ± 1.1e-05        1   0.858375 ± 0.011      -1357.89 ± 360
> SMAC RF              0.132719 ± 0.0012     272.2    0.91347 ± 0.003      -2342.64 ± 180
> DNGO                48.0451   ± 0.47     98521.2   0.858092 ± 0.017      -1018.43 ± 200
> Exact GP             4.48871  ± 0.22      9204.6   0.700453 ± 0.0058      -262.89 ±  20
> SVGP         8.11583  ± 0.11     16642.3   0.879975 ± 0.0037     -1290.43 ±  86
> Vecchia              2.46467  ± 0.063     5054.1   0.758656 ± 0.0065      -468.52 ±  28
> ```
>
> ```
> N=1000, D=100, Sphere (unimodal)
> Surrogate           Fit t (s)       t/t_ENN          NRMSE         LogLik
> ---------------------------------------------------------------------------
> ENN            6.25e-04 ± 1.7e-05         1   1.02681  ± 0.0006  797.65 ± 1
> SMAC RF        1.36649  ± 0.057      2187.6   1.00078  ± 0.0012  825.78 ± 2
> DNGO          75.9242   ± 1.3      121546.1   1.01224  ± 0.0052  800.01 ± 12
> Exact GP     166.773    ± 8.5      266984.3   1.03072  ± 0.0039  774.18 ± 6
> SVGP   9.81392  ± 0.24      15711.0   0.99302  ± 0.00055 802.37 ± 2
> Vecchia        4.07247  ± 0.1        6519.6   1.00168  ± 0.00048 824.64 ± 1
> ```

---

> > ### Author Rebuttal · Reviewer_fWyZ · 2026-04-01
> >
> > Thanks for the rebuttal. However, its hard to raise scores without seeing all the additional results.
> >
> > I don't agree with the statement that you cannot run the other baselines due to time constraints. The entire point of BO is that we have some time to make our decisions. The referenced papers are all relatively fast, perhaps you have not implemnted them well (regardless, Its  not clear that they actually correspond to the methods in your table). In their papers, they definitely tackle large-scale problems.

---

> > > ### Author Response · Authors · 2026-04-02
> > >
> > > > I don't agree with the statement that you cannot run the other baselines due to time constraints.
> > >
> > > Our goal is to demonstrate that replacing GP with ENN in TuRBO results in a dramatic speedup without loss of solution quality. As such, we feel that running egregiously long optimizations on a variety of (slow) methods serves no purpose if we show that (i) TuRBO is a SOTA algorithm, (ii) the running time of an alternative is much longer than TuRBO, and (ii) TuRBO-ENN is much faster than TuRBO.
> > >
> > > This table shows the cumulative proposal time for a single optimization, where we have marked as ">5hr" those cases where the time is 5 hours or more (the same threshold used in the SMAC paper, although we ignore the time spent on function evaluation, similar to Eriksson, et. al (TuRBO) ). We will include this information in Table 1 so that it is clear to the reader why certain runs were omitted from the paper.
> > >
> > >
> > > | opt_name | tlunar:fn | tlunar | push:fn | push | hop:fn | hop | bw-heur:fn | bw-heur |
> > > |---|---|---|---|---|---|---|---|---|
> > > | ucb | 2217.5 | **> 5hr** | **> 5hr** | **> 5hr** | **> 5hr** | **> 5hr** | 10541.8 | **> 5hr** |
> > > | lei | **> 5hr** | **> 5hr** | **> 5hr** | **> 5hr** | **> 5hr** | **> 5hr** | **> 5hr** | **> 5hr** |
> > > | smac | 461.6 | 13031.8 | 15296.1 | 15364.6 | 14648.8 | 17056.4 | 1382.3 | 15630.6 |
> > > | dngo | 364.6 | **> 5hr** | 16893.3 | **> 5hr** | **> 5hr** | **> 5hr** | 2566.3 | **> 5hr** |
> > > | vecchia | 111.5 | **> 5hr** | 4336.6 | **> 5hr** | 4685.2 | **> 5hr** | 491.5 | **> 5hr** |
> > > | ucb:Msparse | **> 5hr** | **> 5hr** | **> 5hr** | **> 5hr** | **> 5hr** | **> 5hr** | **> 5hr** | **> 5hr** |
> > > | turbo-enn | 0.5 | 343.3 | 19.1 | 461.6 | 39.4 | 1402.7 | 1.7 | 543.9 |
> > > | turbo-one | 47.7 | 3119.6 | 946.8 | 3397.6 | 1734.7 | 10595.0 | 145.1 | 3265.9 |
> > >
> > > > The entire point of BO
> > >
> > > While traditionally BO has been focused on expensive-evaluation problems, there has been much work recently on "scalable BO" (BOMO), and this paper contributes to that literature. An important practical application is simulation optimization in engineering, where evaluation time can be small and batch sizes can be large. In this case, it is valuable for the proposal time to be small so as not to dominate (or contribute significantly to) the overall wall time.
> > >
> > > > The referenced papers are all relatively fast,
> > >
> > > Hutter, et. al., (SMAC) use a 5-hour cutoff. Snoek, et. al. (DNGO)  report ~5 min. for a *single* proposal at N=1000 with 32 CPUs. TuRBO reports only a 1min cumulative proposal time for frozen-noise LunarLander (tlunar:fn, above), but they use a GPU, not a CPU (as we do). Each of our evaluations is performed with a single CPU in a container that lasts only for the life of the run.  We use publicly-available author-published reference code for SMAC and Vecchia. We use custom code for TuRBO that is carefully synched to the reference code. (Our code is a little faster than the reference code [mostly refactoring the original into an `ask()`/`tell()` interface] but produces equivalent-quality solutions.) No reference code for DNGO is available, as far as we know, so we make a good-faith effort. We use BoTorch for other methods. All of our code is publicly available.
> > >
> > > Our experiments present an apples-to-apples comparison of all methods, whereas the individual papers you reference seem to vary considerably in their hardware configurations. Additionally, these days GPUs can easily cost 10x-100x per unit time as much as CPUs, and in most organizations CPUs are more plentiful than GPUs, so it is valuable to have computationally-efficient optimization algorithms. All that being said, no matter how cheap or plentiful hardware is, more work can always be done with a more efficient algorithm.

---

### Decision · Program_Chairs · 2026-04-30

**Decision:**

Accept (regular)

**Comment:**

While this paper's scores are quite positive, my view from reading reviewers' actual text is that is that it is very slightly unfinished. Let me summarize its state, based on various comments raised by reviewers, most of whom are seasoned experts.

Strengths
- Well-motivated problem.
- Strong engineering and integration with complex state-of-the-art pipelines such as TuRBO.
- Good performance in the sense of significant computational speedsup while maintaining quality.

Issues
- Missing baselines
- Lack of comparisons to understand what the effects of each individual choice are
- Lack of comparisons of effect of hyperparameter values

On the one hand, I think the strengths are really strong, especially since TuRBO and its derivatives are complicated and not easy to work with, but are necessary when working in settings like the one considered. On the other hand, I think an updated version that actually does the comparisons would be quite stronger. I think, on balance, for me this paper, while slightly unfinished, is good enough - it lands very slightly on the side of accepting now rather than later, largely because the problem class requires methods that are not that easy to work with and yet deserves more work due to its importance. I am happy to be overruled by the SAC or PCs if needed. I very strongly encourage the authors to add the comparisons above to the work, as this would make it substantially more convincing and therefore impactful.